# Boosting mechanical durability under high humidity by bioinspired multisite polymer for high-efficiency flexible perovskite solar cells

Zhihao Li[1,2,3,4], Chunmei Jia[1,2], Zhi Wan[1,2], Junchao Cao[3], Jishan Shi[1,2], Jiayi Xue[1,2], Xirui Liu[5], Hongzhuo Wu[4], Chuanxiao Xiao [5,6], Can Li[1,2], Meng Li [4]✉, Chao Zhang [3]✉ & Zhen Li [1,2]✉

Flexible perovskite solar cells (FPSCs) with high stability in moist air are required for their practical applications. However, the poor mechanical stability under high humidity air remains a critical challenge for flexible perovskite devices. Herein, inspired by the exceptional wet adhesion of mussels via dopamine groups, we propose a multidentate-cross-linking strategy, which combine multibranched structure and adequate dopamine anchor sites in three-dimensional hyperbranched polymer to directly chelate perovskite materials in multiple directions, therefore construct a vertical scaffold across the bulk of perovskite films from the bottom to the top interfaces, intimately bind to the perovskite grains and substrates with a strong adhesion ability, and enhance mechanical durability under high humidity. Consequently, the modified rigid PSCs achieve superior PCE up to 25.92%, while flexible PSCs exhibit a PCE of 24.43% and maintain 94.1% of initial PCE after 10,000 bending cycles with a bending radius of 3 mm under exposed to 65% humidity.

Perovskite solar cells (PSCs) have attracted considerable attention as a promising technology owing to their high flexibility[1]. low weights[2,3]. and mass printability[4]. Through optimizing the perovskite films[5,6]. interfaces[7–9]. and flexible electrodes[10–13]. the certified power conversion efficiency (PCE) of flexible perovskite solar cells (FPSCs) have skyrocketed to 24.90%[14]. Despite this remarkable PCE achievement, their mechanical and environmental stability still cannot meet the demands of practical applications[15,16].

Fracture within the perovskite thin film or at the interfaces is the most significant mechanical failure occurred in FPSCs[17–20]. Fractures in the perovskite thin films often begin at weak points in the films such as grain-boundary or micro-cracks[21]. ultimately leading to performance deterioration in FPSCs during deformation[22]. Meanwhile, the cracks and stress induced lattice distortion could cause unfavorable phase transitions and accelerate degradation of the perovskite film[23–25]. Cracks developed in the film are pathways for moisture infiltration and decomposed species releasing[26]. Huang et al. applied external strains on perovskite films and found that tensile strain accelerates the degradation of perovskites under illumination. Meanwhile, Dauskardt et al. exposed the perovskite films to either damp air (25 °C, 85% RH) or dry heat (85 °C, 25% RH) for 24 h with different stresses[27]. It was found that the tensile stress accelerated degradation of perovskite films and exhibited worst moisture and thermal stability, which was attributed to the lower formation energies of point defects and lower activation

[1]State Key Laboratory of Solidification Processing, Northwestern Polytechnical University, Xi'an, China. [2]School of Materials Science and Engineering, Northwestern Polytechnical University, Xi'an, China. [3]School of Civil Aviation, Northwestern Polytechnical University, Xi'an, Shaanxi, China. [4]Key Laboratory for Special Functional Materials of Ministry of Education, School of Nanoscience and Materials Engineering, Henan University, Kaifeng, China. [5]Ningbo Institute of Materials Technology and Engineering, Chinese Academy of Sciences, Ningbo, Zhejiang Province, China. [6]Ningbo New Materials Testing and Evaluation Center CO., Ltd, Ningbo, Zhejiang Province, China. ✉e-mail: mengli@henu.edu.cn; chaozhang@nwpu.edu.cn; lizhen@nwpu.edu.cn

energies for ion migration. The mechanical-chemical coherent degradation issues are more severe in FPSCs since the flexible polymer substrates are more prone to water penetration[28]. The tensile stresses developed during fabrication and operation could lead to light, heat, and moisture-induced degradation as well as fracture failure in FPSCs, eventually result in loss of device performance[29]. These complex bonding issues highlight the need to improve the mechanical properties of perovskite films even under humid conditions.

Perovskite films prepared by low-temperature solution process contained internal defects at the grain boundary (GB), notably, in the status of bending, stretching, or twisting, GBs are the stress concentration region in the perovskite films[30]. The GBs are more prone to moisture ingression and crack formation, resulting in poor mechanical stability[31]. To address this issue, cross-linking modification[32,33]. dimensional engineering[34] and self-healing GB[35] have been utilized to suppress crack formation and improve the mechanicial robustness of FPSCs. Polymer-based additives can serve *as* adhesives to glue the GB[36]. such as polyvinylpyrrolidone (PVP)[37]. polyvinyl alcohol (PVA)[38]. and polyurethane (PU)[39] can fix water molecules and stitch cracks at the GB regions, thus improving the mechanicial robustness of FPSCs. However, most polymer adhesives need a dry surface and environment to demonstrate the strongest adhesion ability[40]. the solution processed perovskite films inevitably trap polar solvent molecules in the GB and charge transport layer interfaces of the perovskite films[41]. Even worst, the hygroscopic perovskite materials would absorb moisture

from ambient air and further weaken the adhesion ability of the adhesive additives[42].

Inspired by marine mussels that can tightly attach to foreign surfaces in seawater via adhesive proteins (dopamine, DOPA)[43]. introducing the dopamine structure into polymer has become an effective strategy to develop bio-mimic underwater adhesives[44]. Moreover, compared to linear polymers, hyperbranched polymers (HBPs) have a distinctive three-dimensional molecular structure with abundant end groups[45]. This configuration improves the flexibility and deformability of polymer materials due to their nonentangled geometry[46]. The abundant end groups of HBPs can also serve as cross-linkages to improve strength and toughness. Herein, we designed and synthesized a hyperbranched polymer dopamine adhesive (HPDA) with dopamine end groups, as depicted in Fig. 1a. It can intimately bind to the perovskite grains and substrates with a strong adhesion ability, thus enforcing the perovskite grain boundaries and device interfaces. The HPDA self-assemble in the perovskite films to construct a vertical scaffold across the bulk of perovskite films from the bottom to the top interfaces. The oxygen functional groups present in HPDA coordinated with $Pb^{2+}$ and binds the grain boundaries, suppressing the formation and propagation of intergranular cracks. Moreover, the interactions between HPDA and the two charge transport layers $SnO_2$ and spiro-OMeTAD[47]. resulting in stronger interface toughness at the fragile ETL/PVK and HTL/PVK interfaces. Moreover, the dopamine end-groups allow HPDA to achieve strong adhesion in humidity environment,

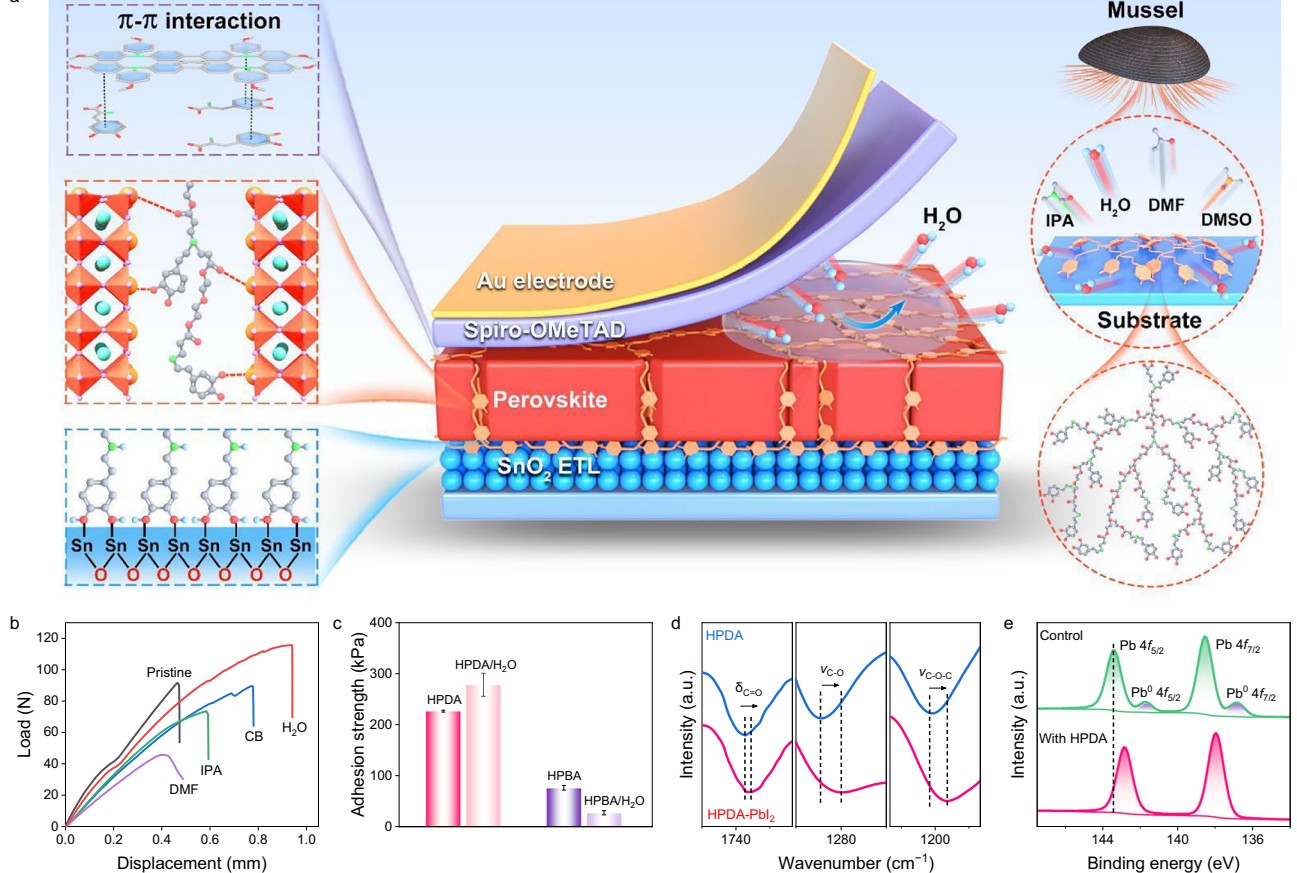

**Fig. 1 | Properties of hyperbranched dopamine polymers (HPDA) adhesive layer. a** Schematics of key component for underwater adhesion feature of natural mussel and HPDA adhesive in perovskite films and interfaces. **b** Lap shear curve of HPDA bonding to ITO substrates under different solvents environment. **c** Adhesion strength of HPDA and HPBA bonding to ITO substrates after washed with water. Error bars represent the standard deviations from the adhesion

strength results of three samples. **d** Fourier-transform infrared (FTIR) spectra of HPDA and HPDA-PbI₂ films. **e** X-ray photoelectron spectra (XPS) of Pb 4f peaks of control and HPDA-modified perovskite films. Source data are provided as a Source data file.

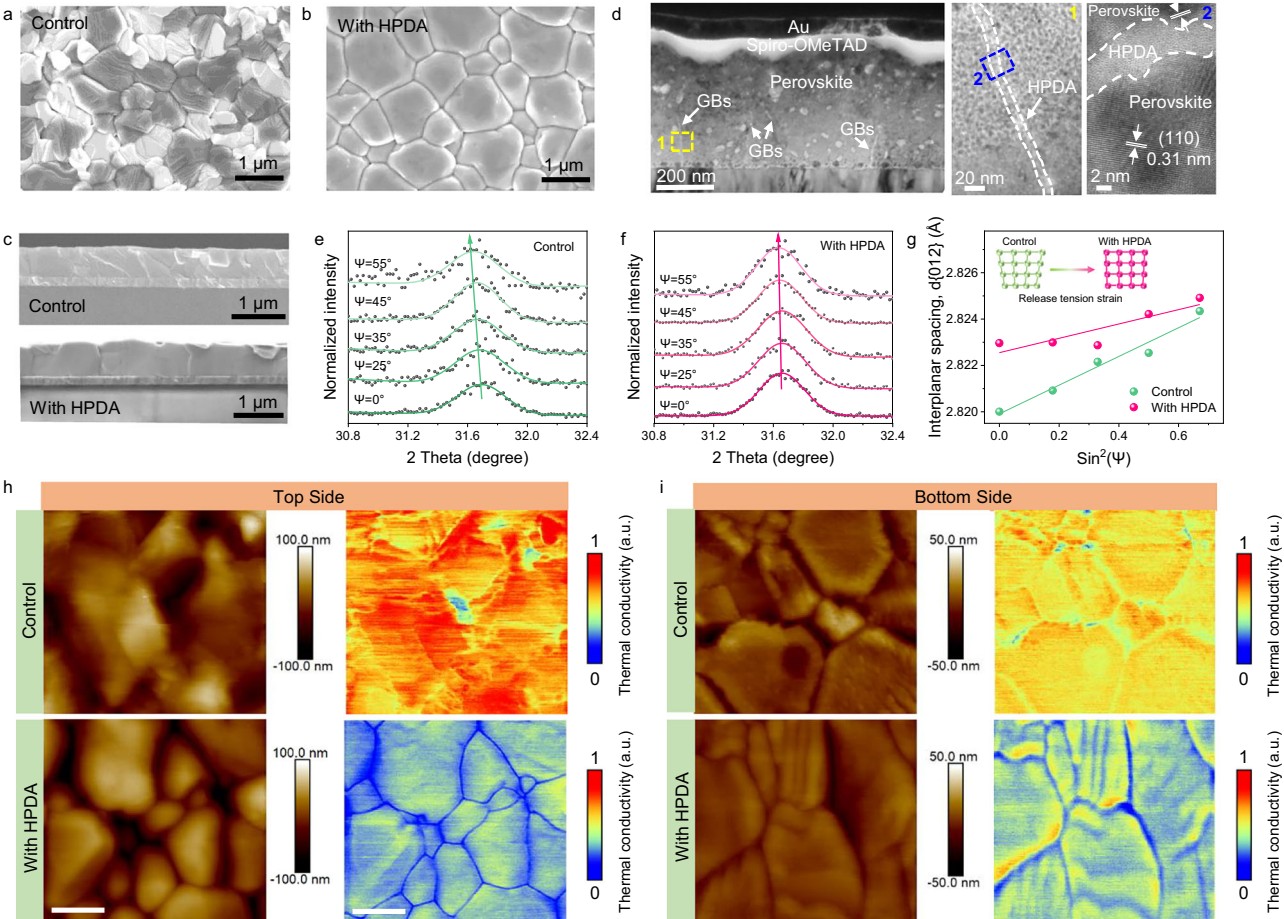

**Fig. 2 | Topography and spatial distribution of the HPDA in the perovskite film.** Top-view of (**a**) control and **b** HPDA-modified perovskite films. **c** Cross-sectional SEM images of control and HPDA-modified perovskite films. **d** Cross-sectional transmission electron microscopy (TEM) images of HPDA-modified PSCs. Grazing incidence X-ray Diffraction (GIXRD) patterns at different Ψ angles (from 0° to 55°) for (**e**) control and **f** HPDA-modified perovskite films, respectively. **g** Lattice spacing d(012) versus sin²(Ψ) plots for control and HPDA-modified perovskite films. SThM topography and corresponding thermal images of top and bottom surfaces for (**h**) control and **i** HPDA-modified perovskite films. Scale bar: 400 nm. Source data are provided as a Source data file.

thereby inhibit the accelerated degradation of FPSCs by cooperative interactions of water and mechanical stress.

Our findings demonstrate that the HPDA-modified rigid PSCs achieve superior PCE up to 25.92%, while flexible PSCs exhibit PCE up to 24.43% and overall improved stabilities with $T_{90} > 22800$ bending cycles, operational stability with $T_{90} > 1140$ h, and ambient stability with $T_{90} > 2690$ h. Moreover, the HPDA can bind tightly to the divalent lead ion to realize a self-encapsulation, substantially mitigating the risk of lead leakage even in the event of structural failure in flexible PSCs.

## Results and discussion
Herein, HPDA was synthesized using Michael addition reaction between multi-vinyl monomers and dopamine hydrochloride, as shown in Supplementary Fig. 1. The molecular structure of the HPDA polymer was verified by ¹H nuclear magnetic resonance (NMR, Supplementary Fig. 2), Fourier transform infrared (FTIR, Supplementary Fig. 3) spectroscopy and gel permeation chromatography (GPC, Supplementary Fig. 4). The HPDA is initially a transparent yellowish liquid, while immediately solidifies into a milk-white, sticky paste upon contacting water (Supplementary Fig. 5). The solvent and moisture resistance of the adhesive was verified through lap-shear tests after the HPDA was adhered to ITO substrates and washed with different solvent (Fig. 1b). In IPA and CB solvent environment, HPDA turned in a swollen state, slightly decreasing the adhesion strength. Meanwhile, although

HPDA had good solubility in DMF, it can still maintain decent adhesion strength. It is interesting that the HPDA demonstrates an excellent adhesive strength to the substrates in water. To illustrate the roles of dopamine end-groups for achieving the strong adhesion underwater, we prepared another HBPs by replacing the dopamine monomer (DOPA) with butylamine (BA), and termed as HPBA (Supplementary Fig. 6). Comparatively, the underwater adhesion strengths of HPBA were significantly decreased compared to those of HPDA (Fig. 1c), which highlighted the importance of the dopamine terminal groups in achieving the strong underwater adhesion[48].

Electrostatic potential of HPDA revealed that the negative charges accumulate on C=O, which provides a favorable position for the coordination with Pb²⁺ cations (Supplementary Fig. 7). The chemical interaction between HPDA and PbI₂ was characterized using FTIR spectroscopy (Fig. 1d). The vibration peaks assigned to −C=O (1733 cm⁻¹), −C-O (1284 cm⁻¹) and −C-O-C- (1202 cm⁻¹) of HPDA moved to the lower wavenumber after the blending with PbI₂, which suggests strong polymer perovskite interaction. The chemical interaction was also characterized using X-ray photoelectron spectroscopy (XPS). The Pb $4f_{5/2}$ and Pb $4f_{7/2}$ peaks (Fig. 1e) shifted to higher binding energy, while the O 1s (Supplementary Fig. 8a) peak moved toward lower binding energy, indicating strong chemical interaction between Pb and O with the presence of electron-rich C=O groups in the HPDA[49]. Furthermore, the Pb⁰ peaks observed in the control film is dismissed after

the incorporation of HPDA, suggesting effective passivation of the uncoordinated $Pb^{2+}$[50]. The C=N and -NH group in perovskite formed hydrogen bonds with the -NH- and -OH groups of HPDA, resulting in a slight shift of N 1$s$ and I 3$d$ peaks in the XPS spectra (Supplementary Fig. 8b, c).

Scanning electron microscopy (SEM) and atomic force microscopy (AFM) measurements were conducted to characterize the influence of HPDA on the morphology of perovskite films. The grain size of HPDA-modified perovskite films is much larger than the pristine films (Fig. 2a, b). Moreover, the cross-sectional images of HPDA-modified perovskite films showed vertical oriented neat grains, while the pristine film showed irregular and smaller grains with numerous grain boundaries (GBs) (Fig. 2c). Due to the strong coordination effect between HPDA and $Pb^{2+}$, HPDA may weaken the coordination interaction between $PbI_2$ and DMF[51]. resulting in larger colloids (Supplementary Fig. 9). The $PbI_2$-HPDA films formed a mesoporous scaffold with large number of pores, which facilitates solution penetration and the reaction between lead halides and ammonium halides (Supplementary Fig. 10). In situ UV−vis spectra of the perovskite films during annealing indicated that the presence of HPDA can form intermediate adducts between organic salt and $PbI_2$, guiding oriented growth of perovskite and achieving larger perovskite grains (Supplementary Fig. 11). The crystallization property of the perovskite was analyzed by X-ray diffraction (XRD) and grazing incidence wide-angle X-ray scattering (GIWAXS). As shown in Supplementary Fig. 12, no obvious peak shift was found in the HPDA-modified perovskite films, suggesting HPDA was not incorporated into the perovskite crystal lattice. The intensity of characteristic (110) peak increased for HPDA-modified films, with the narrower full width at half-maximum as compared to the control film, demonstrating the enhanced crystallinity. Meanwhile, the $PbI_2$ peak at 12.7° disappeared due to the strong interaction between HPDA and $Pb^{2+}$, suggesting the effective passivation of GB defects, contributed to the high quality of perovskite film. Moreover, the azimuthal scan profiles for the (110) facets extracted from the grazing-incidence X-ray diffraction (GIXRD) patterns of the annealed perovskite film indicated HPDA enables the formation of highly ordered perovskite film with the (110) plane parallel to the substrate (Supplementary Fig. 13). As shown in Supplementary Fig. 14, The HPDA-modified films exhibited enhanced absorption, which was consistent with the increased grain size and enhanced crystallinity.

In addition, The HPDA-modified films showed smaller surface roughness with a root-mean-square of 36 nm, as compared to 66 nm of the control film (Supplementary Fig. 15), which was beneficial for good ohmic contact between perovskite and HTL and high photovoltaic performance. The cross-linking of polymer in perovskite was confirmed via transmission electron microscopy (TEM). The TEM characterization was conducted by scraping the perovskite film into powder and dispersing in chlorobenzene with ultra-sonication. As shown in Supplementary Fig. 16, the perovskite grains showed noticeable lattice fringes with a lattice spacing of 0.31 nm, matching to the (110) planes of perovskite crystal. Moreover, an obvious amorphous region was clearly distinguishable between the adjacent grains, indicated HPDA prefer to distribute at perovskite grain boundary[52]. For comparison, the pristine perovskite film was fully crystalline without any amorphous regions at GBs. Cross-sectional high-resolution TEM was used to further characterize the distribution of HDPA along the grain boundaries. As shown in Fig. 2d, the TEM image showed the HPDA cross-linked perovskite film, which can clearly distinguish the grain boundaries (GBs) between the perovskite grains. We further investigated the cross-linking effect of HPDA by exposing the perovskite films to e-beam irradiation and observed the changes within the perovskite GBs. Notably, the rapid volatilization of the organic species under high-energy e-beam caused shrinking of grains and cracking of the GBs[53]. As shown in Supplementary Fig. 17, with the prolongation of the exposure time, perovskite films with HPDA

showed a polymer membrane inside the GBs, which tightly bind the grains together. However, polymer membrane was not observed at the GB of the control films, and the grain boundaries exhibited a significantly faster cracking rate. We infer that the HPDA can stabilize the organic species of perovskite and thus slow down the GBs cracking. The HPDA cross-linking effects within perovskite films increased the fracture toughness of GBs, which leads to greater resistance to tensile stress[54]. Furthermore, the GIXRD measurement was conducted to further confirm the influence of HPDA cross-link the grain boundaries on the residual strain of perovskite lattice. As shown in Fig. 2e, when X-ray incident angle ψ varied from 0° to 55°, the (012) peaks of control films gradually shifted to lower diffraction angles, manifesting larger lattice spacing $d_{(012)}$ as the film depth increase. The changes of lattice spacing indicates a residual tensile lattice strain at the perovskite/$SnO_2$ interface. While the HPDA-modified film showed a negligible shift (Fig. 2f), realizing a released residual stress. As a result, the residual tensile strain of HPDA-modified was 20.36 MPa, constituting ~73% strain reduction compared to 74.68 MPa of the control sample. This result also suggested that HPDA played a critical role in releasing the residual tensile strain of perovskite lattice via strain compensation[55]. Steady-state photoluminescence (PL) and time-resolved photoluminescence (TRPL) measurement were conducted to investigate the quality and charge transport behavior of perovskite with HPDA modification. As shown in Supplementary Fig. 18, the PL intensity of HPDA-modified perovskite films showed obviously enhancement as compared to the control film, accompanied by a blue-shift of the peak positions from 796 nm to 793 nm. Therefore, the HPDA within perovskite can effectively reduce the defects and suppress non-radiative recombination. In addition, the TRPL decay curves were fitted by a bi-exponential decay function: $f(t) = A_1 \exp(-t/\tau_1) + A_2 \exp(-t/\tau_2)$, The fitted parameters were summarized in Supplementary Table 1. As a result, the average charge lifetime ($\tau_{ave}$) was greatly increased from 408 ns (control film) to 757 ns with HPDA treatment (Supplementary Fig. 19), which can be attributed to the enlarged grain with low defect and grain crosslinking by HPDA.

AFM was utilized to probe the phase distribution properties of HPDA on the top and bottom surfaces of perovskite films[56]. A peeling-off process was conducted to expose the bottom surface. As shown in Supplementary Fig. 20, the HPDA-modified perovskite film exhibited obviously different topography and phase images as compared to the control film. For the control film, the AFM phase distribution of the top surface located at −19.8°, and slightly increased to −12.7° for the bottom surface, indicating the chemical composition and mechanical properties of the top surface changed little compared to the bottom surface. For the HPDA-modified perovskite film, the AFM phase distribution of the top surface was 14.4°, combined with barely any signatures of the GB, indicating the perovskite film surface has been uniformly covered with HPDA polymer. In addition, the internal grains of the bottom surface show similar AFM phase as the control perovskite, while the AFM phase of grain boundaries is located at about 12.5°, which was similar to the top surface. This result indicated that the HPDA polymer was deeply inserted into the GBs and reached the top and bottom surfaces, which was also verified by the time-of-flight secondary ions mass spectroscopy (ToF-SIMS) (Supplementary Fig. 21).

Subsequently, AFM force images were employed to quantify the adhesion of the top and bottom of the perovskite films (Supplementary Fig. 22 and Supplementary Table 2). The HPDA-modified perovskite film exhibited obvious enhanced adhesion strength on both the top and bottom surfaces as compared to the control films, once again proved the distribution properties of HPDA. In addition, owing to the existence of HPDA on the bottom surface, the two neighboring hydroxyl groups in catechol can form bidentate hydrogen bonding coordination with $SnO_2$[57]. resulting in the perovskite tightly adhering to the $SnO_2$ ETL, which were confirmed by the XPS and FTIR characterization (Supplementary Fig. 23). Therefore, in the nano-scratch

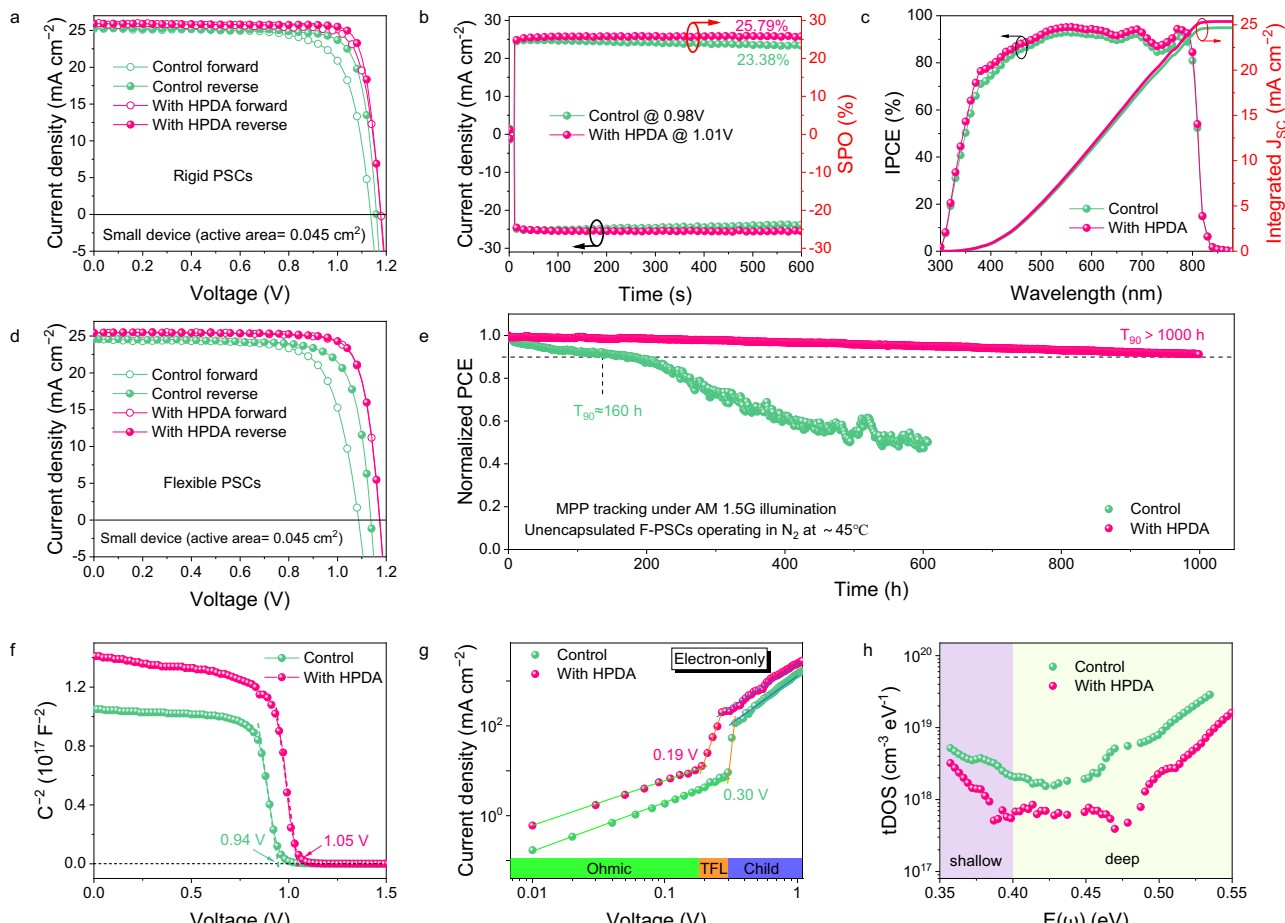

**Fig. 3 | Photovoltaic performance and optoelectronic characterization of PSCs with HPDA modification. a** *J–V* curves of the champion devices with and without HPDA. **b** Stabilized photocurrent and power output of PSCs. **c** External quantum efficiency (EQE) spectra and integrated current of the PSCs. **d** *J–V* curves of champion flexible devices with and without HPDA. **e** Continuous maximum power point (MPP) tracking of unencapsulated FPSCs with and without HPDA modification in nitrogen atmosphere. **f** Mott–Schottky plots of PSCs with and without HPDA modification. **g** SCLC measurements of control and HPDA-modified PSCs with the electron-only structures. **h** tDOS of PSCs with and without HPDA modification. Source data are provided as a Source data file.

test of the samples, the HPDA-modified film had a larger adhesion strength than the control film (Supplementary Fig. 24). Meanwhile, the HPDA-modified perovskite film exhibited obvious enhanced adhesion strength on the top surface, and the benzene ring in the DOPA structure can also generate π-π interactions with Spiro-OMeTAD, which were verified by the XPS and FTIR characterization (Supplementary Fig. 25). Therefore, the fracture energy of perovskite and Spiro-MeOTAD was measured using the double-cantilever-beam (DCB) method, and the corresponding $G_C$ of HPDA-modified sample was higher at 1.30 J·m$^{-2}$ than 1.02 J·m$^{-2}$ of control device. (Supplementary Fig. 26). The unique adhesive layer can be viewed as an indivisible part of the perovskite absorber film, tailoring the electronical and mechanical properties of the interface between perovskite and transport layers.

Scanning thermal microscopy (SThM) was applied to further characterize the distribution properties of HPDA within perovskite (Fig. 2h, i). In most of the cases, polymers were usually considered as insulators and presented low bulk thermal conductivity[58]. The SThM carried out thermal scans at the top and bottom surfaces of perovskite films and observed low thermal conductivities at grain boundaries within HPDA-modified perovskite film, indicating HPDA mainly distributed at GBs, which was consistent with the aforementioned results. Thus, our work clearly showed the spatial distribution of HPDA polymer in the perovskite film.

Subsequently, we fabricated PSCs with a structure of ITO/SnO$_2$/ Perovskite (without or with HPDA)/Spiro-OMeTAD/Au to investigate the effect of HPDA on the device performance. The optimized device performance with different concentrations of HPDA was presented in Supplementary Fig. 27 and Supplementary Table 3. Figure 3a depicted the *J–V* curves of champion devices and corresponding photovoltaic parameters were given in Supplementary Table 4. The champion PCE of control device was 23.72%, with an open-circuit voltage ($V_{OC}$) of 1.160 V, short-circuit current density ($J_{SC}$) of 25.33 mA cm$^{-2}$ and fill factor (FF) of 80%. In contrast, the optimized HPDA-modified device exhibited high PCE of 25.92% ($V_{OC}$ = 1.178 V, $J_{SC}$ = 25.82 mA cm$^{-2}$, FF = 85%). Supplementary Fig. 28 showed that the HPDA-modified devices had narrow distribution with satisfactory parameters, confirming the enhanced PCE and excellent reproducibility. The PCEs from *J–V* curves were proved by the stabilized output power efficiencies of 23.38% and 25.79% for control and HPDA-modified devices, respectively (Fig. 3b), with the integrated $J_{SC}$ via external quantum efficiency (EQE) was 24.67 mA cm$^{-2}$ and 25.37 mA cm$^{-2}$ (Fig. 3c), matching well with the $J_{SC}$ obtained from *J–V* results. We further use 1-step antisolvent method to prepare inverted PSCs and obtain a PCE of 25.54%, demonstrating that incorporating the HPDA polymer into the perovskite is a universal method (Supplementary Fig. 29). Meanwhile, we fabricated large–area (1 cm$^2$) device with champion PCE of 24.68% for HPDA-modified PSCs (Supplementary Fig. 30), demonstrating the additive strategy was

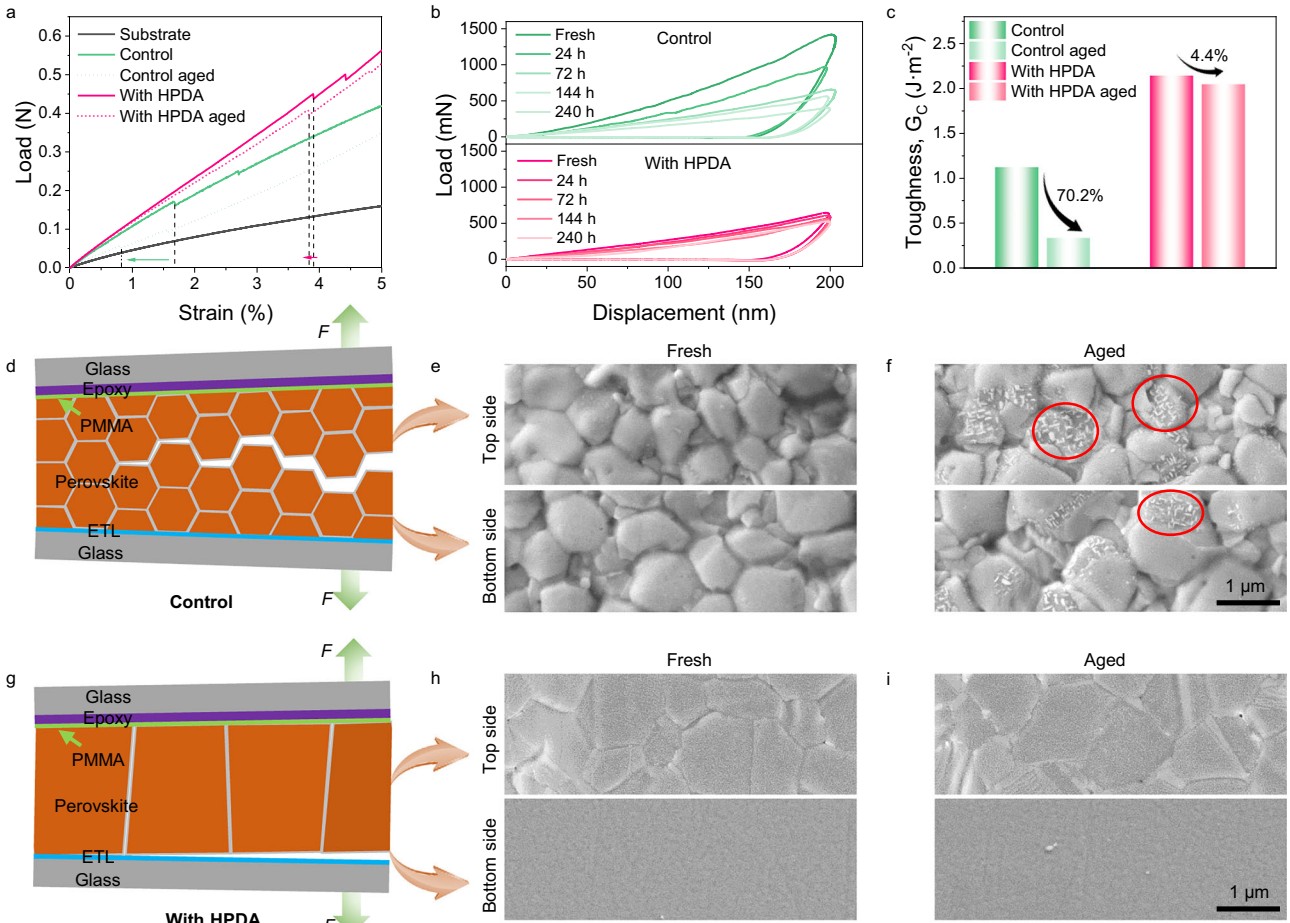

**Fig. 4 | Influence of humidity on perovskite mechanical stability. a** Load–strain curves of the substrate, control, and HPDA-modified perovskite films. **b** Representative load-displacement curve of indentation test for control and HPDA-modified perovskite film after exposing to ambient humidity. **c** Fracture energy of PSCs with and without HPDA modification. **d** Schematic intergranular fracture. **e**, **f** SEM images of fracture surface of the fresh and aged control PSCs. **g** Schematic interface fracture. **h**, **i** SEM images of fracture surface of the fresh and aged HPDA-modified PSCs. Source data are provided as a Source data file.

feasible for larger area PSCs. The planar FPSCs were also fabricated on PET substrate with the same structure of rigid PSCs. HPDA-modified FPSCs demonstrated a high PCE of 24.43% (Fig. 3d and Supplementary Table 5) with complete statistic summary of parameters given in Supplementary Fig. 31 and Supplementary Table 6. The SPO was 23.55% (Supplementary Fig. 32) and integrated $J_{SC}$ was 25.00 mA cm$^{-2}$ (Supplementary Fig 33), agreeing well with the measured values obtained from $J-V$ results. FPSCs with 1 cm$^2$ possessed a high PCE of 23.01% (Supplementary Fig. 34). The operational stability was tested at maximum power point under continuous illumination in N$_2$ atmosphere. The HPDA-modified PSCs remained 90% of its original efficiency after 1000 h, while the PCE of control device declined considerably (Fig. 3e). This excellent long-term stability can be attributed to the all-round passivation of grain boundaries and interface by HPDA.

Subsequently, we conducted the dependency of $V_{OC}$ on light intensity to investigate the recombination characteristics. As shown in Supplementary Fig. 35, the control device exhibited a slope of 1.46 kT/ q, while the slope decreased to 1.13 kT/q for HPDA-modified devices, indicating the trap-assisted recombination was effectively suppressed. To further investigate the effect of HPDA on energy levels of perovskite film, we calculated the Fermi level (WF), valence band (VB), and conduction band (CB) from ultraviolet photoelectron spectroscopy (UPS, Supplementary Fig. 36a) and Tauc plot (Supplementary Fig. 36b), the corresponding values were listed in Supplementary Table 7. The energy level diagrams of PSCs were shown in

Supplementary Fig. 36c. The VBM of control and HPDA-modified perovskite were calculated to be −5.76, and −5.82 eV, respectively. The deeper VBM of HPDA-modified perovskite contributed to higher open-circuit voltage ($V_{OC}$) of devices. In addition, the built-in potential ($V_{bi}$) obtained from Mott-Schottky analysis increased from 0.9 V for control to 1.02 V for HPDA-modified device (Fig. 3f). The improved $V_{bi}$ can provide effective driving force for charge separation, resulting in the higher $V_{OC}$. Due to the external thermal and mechanical stress, FPSCs had more defects relative to rigid devices and resulting in the increased $V_{OC}$ loss and reduced FF. Therefore, the defect density of FPSCs was investigated. The HPDA-modified PSCs exhibited reduced trap density via space-charge-limited-current (SCLC) measurement (Fig. 3g). The small dark current leakage also demonstrated that the perovskite film with HPDA had fewer defects (Supplementary Fig. 37). The trap density of states (tDOS) was further analyzed by thermal admittance spectroscopy (Fig. 3h). The shallower traps (0.35–0.4 eV) corresponded to grain boundary and the deeper traps (0.4–0.52 eV) originated from surface were all reduced after HPDA modification, indicating that the HPDA can effectively passivate defects[59].

The presence of environment stimuli during the operation of PSCs will not only cause wholesale chemical degradation of perovskite materials and interfaces, but also affect mechanical behavior. Moisture penetration into GBs can lead to rapid degradation of perovskite films. Therefore, we investigated the effects of HPDA modification on mechanical properties under humidity aging. Figure 4a showed the

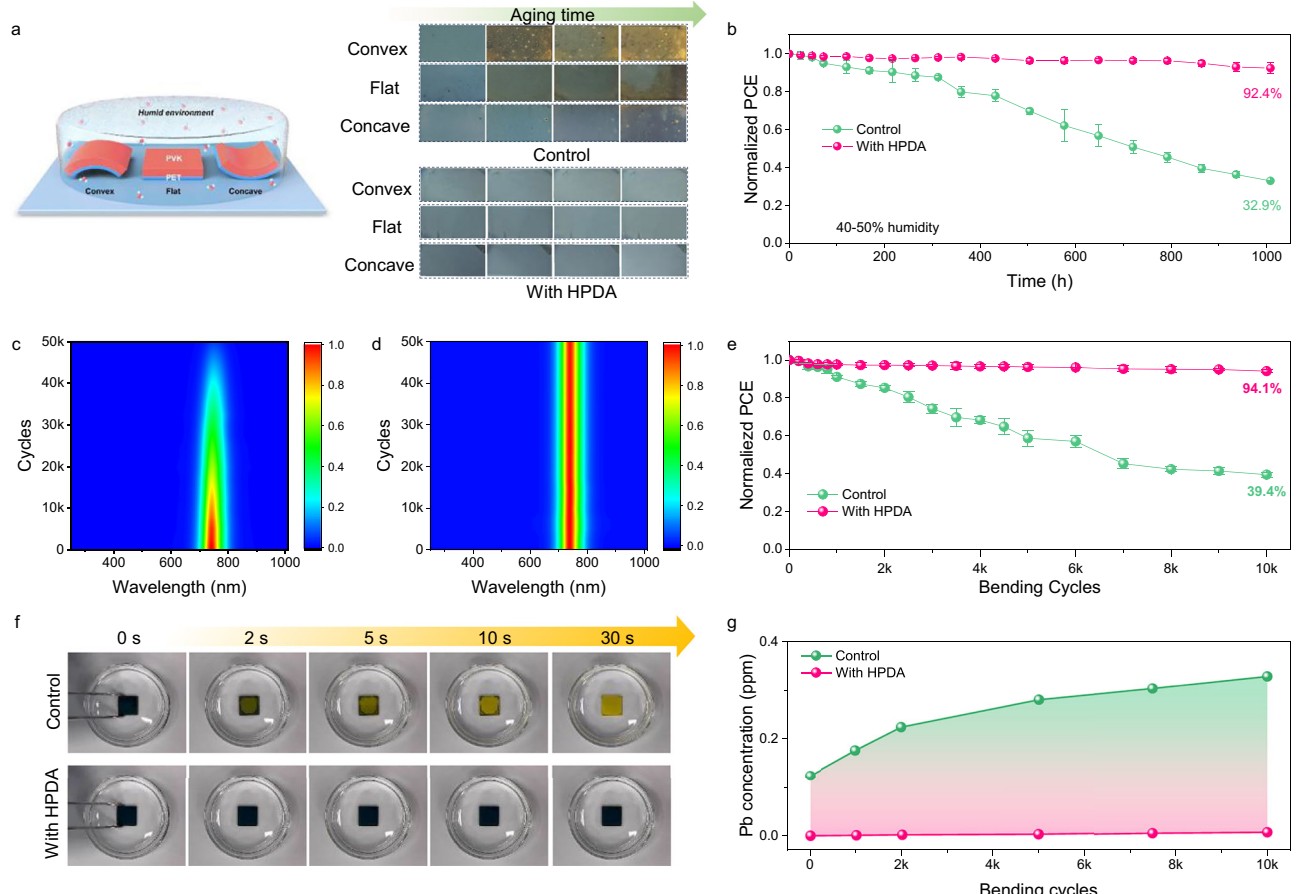

**Fig. 5 | Influence of strain on perovskite film humidity stability. a** Schematic diagram of the experimental setup and corresponding photographs of the films with different strains after 120 h of aging at 25 °C and 85% RH. **b** Environmental stability of unencapsulated FPSCs with and without HPDA modification by bending in a convex shape and exposed to 40−50% RH. Error bars represent the standard deviations from the statistic results of two devices. Comparison of in-situ PL spectra of (**c**) control and **d** HPDA-modified perovskite films with 50,000 bending cycles in 85% humidity. **e** Normalized PCE for FPSCs as a function of bending cycles with a bending radius of 3 mm. Error bars represent the standard deviations from the statistic results of two devices. **f** Photograph evolution of control and HPDA-modified perovskite film when dripping into the water. **g** Pb concentration in water measured by ICP-MS. The perovskite films bended with different cycles were washed with water for 10 min. Source data are provided as a Source data file.

load−strain curves of perovskite samples and the dashed lines correspond to the strain at the first rupture of the perovskite films. The fracture strength of the HPDA-modified film was 43.73 MPa, which was much higher than that of the control film of 13.21 MPa, and the elongation at break also increased from 1.68% to 3.92%. Meanwhile, the HPDA-modified perovskite film exhibited smaller changes of the fracture strength after aging from 43.73 to 39.32 (Δ = 10%) compared to the control device (from 13.21 to 2.67, Δ = 79%). The reduced fracture strength change indicated that the hyperbranched network topology structure of HPDA can simultaneously realize the excellent strength and toughness of perovskite film.

When exposed to humid environment, obvious morphological changes were observed in SEM (Supplementary Fig. 38). It can be expected that the morphological changes on the perovskite surfaces will significantly affect the mechanical properties of the material. In particular, the occurrence of crack propagation can be regarded as a sign of the gradually reduced hardness of the material with the increase of water content. In order to further investigate this matter, the local mechanical properties of perovskite films were quantitatively characterized with the nano-indentation measurement by exposing the perovskite film with different time in humid conditions. The behavior of the Young's modulus as a function of time was shown in Fig. 4b. It clearly shown that the Young's modulus decreased steadily in control film, while the HPDA-modified film exhibited minor changes. The retarded moisture tolerance ascribed from HPDA located at GBs and surface of perovskite was confirmed in detail.

To further study the role of HPDA modification on perovskite, we measured the fracture energy (Gc) using the DCB measurements, as illustrated in Supplementary Fig. 39. The HPDA modified perovskite exhibited a larger fracture energy of 2.14 J m⁻² compared to 1.12 J m⁻² of the control sample (Fig. 4c). The mating fracture surfaces of perovskite samples after DCB measurement were also investigated in the SEM. As shown in Fig. 4e, the perovskite not only existed on the top half of the control fracture surface, but also presented on the mating fracture surface on the bottom side. This indicated that fracture in the control sample occurred within the perovskite film, indicating intergranular fracture (Fig. 4d). Figure 4h was SEM image of the fracture surface of the top half of the HPDA-modified samples, presenting a smooth fracture surface. Moreover, there was barely presence of perovskite on the bottom side of the failed DCB sample. These results indicated that the failure had occurred at the PVK/ETL interface, as depicted schematically in Fig. 4g. Compared to the control films, there was barely horizontal lower-toughness grain boundaries in the HPDA-modified films, thereby forcing the fracture to occur along the next weakest path, i.e., the PVK/ETL interface. Owing to the existence of HPDA on bottom surface, the perovskite can tightly adhere to the SnO₂ ETL,

resulting in the enhancement of Gc compared to the control sample as previously reported. Meanwhile, the control film showed some white particulate, which marked with red circle, indicating the decomposition process (Fig. 4f). In contrast, the HPDA-modified sample exhibited barely change (Fig. 4i). Moreover, the HPDA-modified perovskite film exhibited smaller changes of the Gc after aging compared to the control device. This supported the above argument that HPDA can crosslink adjacent grains and prevent moisture infiltrating into perovskite crystal through GBs, thus, maintaining excellent mechanical properties in humidity environment.

To observe the impact of stress on film stability under humidity, we adjusted the stress of perovskite films on PET substrates by applying quasi-static loading with bending in a convex shape (tensile stress) or in a concave shape (compressive stress)[60]. which was illustrated in the inset of Fig. 5a. The samples were exposed to 85% ambient humidity and the control film with applied tensile stress had significant color change from black to yellow, indicating the degradation of perovskite with $PbI_2$ formation, whereas the films with applied compressive stress remained mostly black with a little yellowish after exposure. For perovskites, bending in a convex shape not only leads to a tensile stress, but causes microcracks, which will create an accelerated pathway for the intrusion of moisture, resulting in the degradation of perovskite[61]. In contrast, the HPDA-modified perovskite film keeps integral morphology in both cases. This observation provides evidence of the importance of HPDA played in the degradation of perovskite due to moisture and the stress of the film. Notably, the long-term stability of FPSCs was studied by bending in a convex shape and exposed to 40–50% RH at room temperature for 1000 h, shown in Fig. 5b. The HPDA-modified device was stable of maintaining 92.4% initial PCE, which suggested the compactness of the film and strong hydrophobic of HPDA contributed to achieve an admirable humidity stability even under tensile stress (Supplementary Fig. 40).

To further explore the effect of stress build-up on perovskite humidity stability, we conducted fatigue tests by applied the stress under high humidity, as a result of cyclic loading by bending in a convex shape. Firstly, we conducted in situ PL measurements under 85% ambient humidity (Fig. 5c, d). It was found that the PL intensity of the control film decreased with increasing bending cycles, and significantly declined after 20,000 times. In contrast, the PL intensity of the HPDA-modified perovskite film had almost undetectable change. The improvement of bending resistance to humidity originates from the improved toughness and water resistance of the HPDA-modified perovskite film. In addition, we carried out fatigue tests with a curvature radius of 3 mm and 10,000 bending cycles under 65% ambient humidity (Fig. 5e). With the introduction of HPDA, FPSCs retained 94.1% of their original PCE, while the control device retained only 39.4% of its initial PCE. Moreover, finite element simulation analysis indicated that the introduction of HPDA layer can significantly reduce the stress of the perovskite layers, which could improve the mechanical durability of the device (Supplementary Fig. 41). To investigate the impact of using HPDA as internal encapsulation for mitigating lead leakage[62]. we firstly immersed the perovskite films in water for several time, and found the HPDA-modified film could maintain a black phase, while the control film changed to a yellow phase while approaching water (Fig. 5f). Therefore, the HPDA can reached 99% lead sequestration efficiency compared to the control device (Supplementary Fig. 42), due to the high adsorption rate (5.75 mg min$^{-1}$ g$^{-1}$) and saturated adsorption capacity (56.49 mg g$^{-1}$) of HPDA (Supplementary Fig. 43). For perovskite, stress build-up leads to further expansion of cracks, which creates an accelerated path for the ingress of moisture, leading to degradation of perovskite and further lead leakage, resulting in the environmental pollution. The perovskite films were first bending for different cycles and then water dripped. The control perovskite film showed increased cracks (Supplementary Fig. 44) and significantly elevated Pb concentration with the growth of bending

cycles, while the HPDA-modified sample showed negligible change (Fig. 5g). The strong lead capturing capability of HPDA was also proven by the results of FPSCs bent with different directions (parallel, vertical, and diagonal to the electrode direction) and crumpled (Supplementary Fig. 45a). In addition, 99% of lead leakage was prevented by incorporating HPDA modification went through acidic rainfall test and hot water soaking test (Supplementary Fig. 45c, d). All tests confirm that HPDA can effectively prevent lead leakage in flexible PSCs. Therefore, the HPDA with strong capability with $Pb^{2+}$ can help to prevent the environmental pollution and expected to facilitated the future wearable application of FPSCs.

In summary, we designed and synthesized a hyperbranched dopamine polymer adhesive (HPDA) with a hydrophobic backbone and hydrophilic adhesive catechol side branches inspired by mussels. HPDA can coordinate with $Pb^{2+}$ to crosslink the 3D perovskite grain boundaries. Moreover, the two neighboring hydroxyl groups in catechol formed bidentate hydrogen coordination with $SnO_2$, the benzene ring in the DOPA structure can also generate π-π interactions with Spiro-OMeTAD, resulting in strong adhesion at the fragile ETL/PVK and HTL/PVK interface. Therefore, the HPDA were shown to form a stereoscopic network from the interface to the bulk perovskite in the whole device with toughness and water resistance properties. As a result, the HPDA-modified device exhibited champion PCE of 25.92% and 24.43% on rigid and flexible substrates, respectively. The HPDA-modified flexible devices exhibited simultaneous improvements of fracture toughness in perovskite film, ETL/PVK and HTL/PVK interface. This leads to robust FPSCs that maintain 94.1% of their initial PCE after 10,000 bending cycles with a bending radius of 3 mm under exposed to 65% humidity. This work provides a comprehensive and multi-dimension fracture toughness reinforcement strategy for highly stable flexible perovskite photovoltaics.

## Methods

### Materials

Glass/ITO (8 Ω/sq) and PET/ITO (15 Ω/sq) were purchased from Advanced Election Technology Co., Ltd and YingKou Libra New Energy Technology Co., Ltd., respectively. $SnO_2$ colloidal dispersion (15% in $H_2O$) were purchased from Alfa Aesar. Pentaerythritol tetra-acrylate (PETEA, 99%), poly(ethylene glycol) diacrylate (PEG-200) were purchased from TCI (Shanghai, China). Dopamine hydrochloride (DOPA, 99%, Sigma-Aldrich, Shanghai, China), triethylamine (TEA, 99%, Heowns China), methyl tertiary butyl ether (MTBE, 98%) were used as received. Methylammonium chloride (MACl, 98%), formamidine iodide (FAI, 99.5%) and methylammonium bromide (MABr, 99.5%) were purchased from Great Cell Solar Corp. Lead (II) iodide ($PbI_2$, 99.99%) was purchased from TCI Shanghai Chemical Industry Materials Corp. Dimethyl formamide (DMF, 99.8%, anhydrous), dimethyl sulfoxide (DMSO, 99.9%, anhydrous), chlorobenzene(CB, 99.8%, anhydrous), 2-propanol (IPA, anhydrous, 99.5%) and acetonitrile (ACN, 99.8%, anhydrous) were purchased from Sigma-Aldrich without further purification. Methanol was purchased from Sinopharm Chemical Reagent Co., Ltd. 2,2',7,7'-tetrakis(N,N-di-p-methoxyphenylamine)-9,9-spirobifluorene (Spiro-OMeTAD, ≥99.5%), bis(trifluoromethanesulfonyl)imide (Li-TFSI, ≥99%) and 4-tert-butyl pyridine (tBP, ≥96%) were purchased from Xi'an Polymer Light Technology Corp.

### Materials synthesis

The hyperbranched polymer was synthesized via one-step of Michael addition reaction in a weak alkaline medium. In brief, PETEA (2.64 g 7.5 mmol), PEGDA-200 (3 g 15 mmol), DOPA (6.5 g 34.2 mmol) and DMSO (40 mL) were simultaneously added into an around-bottomed flask. The molar ratio of -C=C to $-NH_2$ was 1.8:1. The mixture was stirred until clarification, and then TEA was added dropwise to adjust the pH to 8. The reaction was kept in the oil bath at 80 °C with stirring for 3 h

in the dark. After completion of polymerization, the product was collected by suction filtration to remove salts generated during the reaction. Then the clear solution was purified with five-fold precipitation agent (MTBE) for three times. Finally, the HBP was steamed to remove residual solvent and precipitation agent.

## Device fabrication

Glass/ITO and PET/ITO substrates were cleaned in an ultrasonic bath with the procedure of detegent, acetone and ethanol for 15 min each, and then the substrates were treated with UV-Ozone for 20 min to improve the hydrophilicity. For fabricating the $SnO_2$ layer, the $SnO_2$ solution was diluted with DI water (with 1:6 volume ratio) and sonicated for 30 min. The $SnO_2$ solution was spin-coated on ITO at 4000 rpm for 30 s and annealed at 150 °C for 30 min. $(FAPbI_3)_{1-x}(MAPbBr_3)_x$ perovskite were prepared through two-step deposition methods. Firstly, 1.5 M $PbI_2$ dissolved in a DMF-DMSO (9:1 volume ratio) mixed solvent was spin-coated onto $SnO_2$ ETL at 1500 rpm for 30 s, followed by annealing at 70 °C for 1 min. For HPDA-modified sample, 0.02 mg HPDA was added to the $PbI_2$ solution. Subsequently, an FAI:MABr:MACl (90:9:9 mg in 1 mL IPA) solution was spin-coated onto the $PbI_2$ surface at 2000 rpm for 30 s, followed by annealing at 150 °C for 15 min in ambient air (≈40% humidity). Finally, the spiro-OMeTAD hole transport layer was deposited on perovskite film at 3000 rpm for 30 s. The HTL solution contained 90 mg of spiro-OMeTAD doped with 28.8 μL of 4-tert-butylpyridine (tBP) and 17.5 μL of lithium bis(trifluoromethylsulphonyl)imide acetonitrile solution (LiTFSI, 520 mg/mL) in 1 mL of chlorobenzene. Finally, 100 nm Au was deposited on HTLs through thermal evaporation.

## Characterization

The morphologies of perovskite films, fracture surfaces, and the cross-sectional of PSCs were investigated by a field emission scanning electron microscopy (JSM 6700F). The distribution of HDPA was characterized by high-resolution TEM with a FEI Talos F200X operating on 200 kV electron gun. The Surface morphology of the ETLs was characterized using AFM (Bruker, Dimension Fastscan). Fourier transform infrared (FTIR) spectra were recorded in ATR mode using a infrared spectroscope (Thermo, Nicolet 6700). $^1$H-NMR spectra were measured with a Bruker AVANCE III 600 instrument operating at 500 MHz at 298 K. The crystal structure of perovskite were characterized using XRD (Bruker AXS, D8 Advance) and grazing-incidence wide-angle X-ray scattering (GIWAXS, BL14B1 beamline of the Shanghai Synchrotron Radiation Facility (SSRF) using X-ray with a wavelength of 0.6887 Å). In-situ UV–vis spectra of perovskite intermediate phases film are obtained using a homemade system with a fiber optic spectrometer (Ocean Optics, QE Pro). The XPS and UPS was performed using Shimadzu Krato s photoelectron spectrometer. The time-of-flight secondary-ion mass spectrometry (ToF-SIMS) spectra was measured with TOF SIMS5, ION-TOF GmbH. Steady-state photoluminescence spectroscopy (PL) measurements were acquired using a homemade system with a fiber optic spectrometer (Ocean Optics, QE Pro). The time resolved photoluminescence (TRPL) spectra were acquired using an Edinburgh Instruments FLS920. The J–V curve was measured using a solar simulator (Newport 94063 A Oriel Sol3A, Class AAA) under AM 1.5 G (100 W cm$^{-2}$) illumination with light intensity calibrated by a KG-5 filtered silicon standard solar cell (SRC-2020, NREL calibrated), and the J–V data were recorded with a digital source meter (Keithley 2400) with scan speed of 0.1 V s$^{-1}$ for both reverse and forward direction. To ensure the accuracy of the $J_{SC}$ measured from J–V scans, a mask with an aperture area of 0.045 cm$^2$ was covered during the measurement. The incident photocurrent conversion efficiency spectra were measured by using a QE-R 3011 system (Enli Tech, Taiwan). Mott-Schottky analysis was carried out on an electrochemical workstation (Chenhua 760) with the 1 kHz AC frequency and 0–1.5 V bias. Mott-Schottky analysis was carried out on an electrochemical workstation (Chenhua 760) and

was measured in the 0–1.3 V voltage rang and 1000 Hz frequency under dark.

## FIB tomography for cross-sectional PSCs

The cross-sectional PSC lamella was FIB milled using an FEI Helios G4 CX Dual-beam FIB/SEM. With the electron beam, using secondary or backscattered electrons to identify the areas that need to be tested. Then with the ion beam a PSC lamella of that area is cut and transferred onto a special grid where the lamella is attached with platinum deposition. On the grid, the lamella is further thinned in the ion beam using an FEI Helios G4 CX Dual-beam FIB/SEM. until it is electron transparent. The grid is transferred from the FIB-SEM to a TEM to analyse the lamella at high resolution.

## Scanning Thermal Microscopy (SThM) measurements

In SThM measurement, an effective Joule heating can be generated in the probe by a larger AC/DC current going along the probe. The heat flow goes from probe to the perovskite film, the temperature of the probe changes depending on the thermal properties of the perovskite films, which can be used for thermal property measurement of perovskite.

## In situ PL stability tests

The bending durability of f-PSCs was evaluated using a mechanical tester (PR-BDM-100, Puri, China) in constant-radius bending mode with a bending radius of 3 mm in an 85% humidity environment. A homemade system equipped with a fiber optic spectrometer (Ocean Optics, QE Pro) was positioned above the perovskite sample to record the PL spectra at various bending intervals. The corresponding PL spectra were periodically measured in the flat state to ensure consistent positioning during the test.

## Lap shear tests

HPDA was placed between two identical substrates and held by clips. The lap shear adhesive test was carried out after the adhered substrates were washed with different solvents. Lap shear testing was used to evaluate the adhesion stenghths, which was performed on a universal testing machine (CMT 5305). Specimens were pulled apart at a speed of 5 mm/min at 25 °C. The force was measured via a 250 N capacity load cell. The displacement was measured by displacement sensor in the universal testing machine.

## Tensile tests of perovskite films

The substrate is PDMS/PH1000 and the dimensions are 1.8 cm long and 1 cm wide. The thickness of PDMS and perovskite film are 150 μm and 400 nm, respectively. The stretching rate was 2.1 mm min$^{-1}$. Considering that the modulus of PH1000 is close to that of PDMS and the thickness is thin (~20 nm), we ignore the effect of interfacial interactions by PDMS/PH1000 on the strain of perovskite films. The fracture strength was determined using the following expression.

$$\delta = \frac{F - f}{S}$$

Where δ is fracture strength, F is the loading force when the perovskite films break on the substrate, f is the load force for the elasticity substrate, S is the cross-sectional area of the sample. The test structure is as follows.

## Nano scratch measuremnts

In the nanoscratch test, the diamond indenter was used to scratch the control and HPDA-modified perovskite films, while gradually increasing the load on the needle. The load force applied at the moment when the coating is completely scratched through or when there is obvious peeling of the coating is the bonding force of the coating.

## Double-cantilever beam (DCB) measurements

For the PVK fracture energy measurement, the "sandwich" double-cantilever beam (DCB) specimens were prepared with the following structure: glass/ITO/SnO$_2$/PVK/PMMA/epoxy/ITO glass. The dimension of the ITO coated glass substrate used is $37.5 \times 12.5 \times 1$ mm$^3$. The deposition of SnO$_2$ and PVK followed the afore-mentioned procedure. Before bonding, PMMA layer (~800 nm) was spin-coated onto the perovskite films as a barrier to the epoxy. Subsequently, a thin layer of epoxy (~2 μm) was applied onto the PMMA layer to "glue" another cleaned ITO-coated glass substrate on top and cured overnight at room temperature in a glovebox. Excess epoxy was cleaned from the edges of the specimen before mechanical testing was conducted.

For the PVK/HTL interface fracture energy measurement, DCB specimens were prepared with the following structure: glass/ITO/SnO$_2$/PVK/Spiro-OMeTAD/Au/epoxy/ITO/glass. After Au deposition process, a glass substrate of $37.5 \times 12.5 \times 1$ mm$^3$ was attached to the previously sample using epoxy to make a sandwiched structure.

## Bending durability tests

The bending durability of FPSCs was evaluated using a mechanical tester (PR-BDM-100, Puri, China) in constant-radius bending mode with a bending radius of 3 mm in air. The corresponding J–V curves were periodically measured at flat state under AM 1.5 G 100 mW cm$^{-2}$ illumination.

## Pb leakage measurements

In order to investigate the impact of using HPDA as internal encapsulation for mitigating lead leakage under rain, we encapsulated the top side and edges of the FPSCs with 10-μm-thick PET tapes and intentional introduced a $5 \times 5$ mm$^2$ holes. Subsequently, we employed a homemade device to simulate the downpour weather and analyzed the lead concentration via inductively coupled plasma mass spectrometry (ICP-MS).

## Reporting summary

Further information on research design is available in the Nature Portfolio Reporting Summary linked to this article.

## Data availability

Source data are provided with this paper.

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

## Acknowledgements

Zhen L. acknowledges the National Natural Science Foundation of China (52172238, 52102304, 52302327), Open project of Shaanxi Laboratory of Aerospace Power (2021SXSYS-01-03), and the Fundamental Research Funds for the Central Universities (3102019JC0005). C.Z. acknowledges the Fundamental Research Funds for the Central Universities (5000220118). We also thank the Analytical & Testing Center of NWPU for the supports on material characterizations.

## Author contributions

Zhihao L. and Zhen L. conceived the original idea and prepared the maunscript. Zhihao L., C.J., and Z.W. carried out the fabrication, optimization and characterization of the perovskite solar cells. J.C. carried out the mechanical tests under the supervision of C.Z. J.X. carried out the finite-element simulation. X. L. and C. X. carried out the scanning thermal microscopy characterization. H.W., J.S., and C.L. assisted in the schematic drawing. Zhihao L., M.L., C.Z., and Zhen L. participated in the data analysis and result discussions. All authors have given approval to the paper.

## Competing interests

The authors declare no competing interests.
