## [Transparent Peer Review file · Nature Communications]

Boosting Mechanical Durability under High Humidity by Bioinspired Multisite Polymer for High-Efficiency Flexible Perovskite Solar Cells

Corresponding Author: Professor Zhen Li

Version 0:

Reviewer comments:

Reviewer #1

(Remarks to the Author)

The authors have incorporated an innovative HPDA with strong adhesion properties to perovskite layers and interfaces, resulting in significant enhancements in mechanical stability under humid conditions. The use of this material group in FPSCs is noteworthy. However, the explanation of the underlying mechanisms is lacking, and certain results require additional validation. The detailed comments are as follows:

1. Mechanism of Underwater Adhesion:

In Figure S5, it is crucial to provide a detailed explanation of the chemical reactions or interactions responsible for the water-triggered coacervation of HPDA adhesives. It is necessary to determine whether coacervation could occur in the perovskite film under ambient conditions and potentially impair charge transport at the grain boundaries.

In Figures 1b and S6, further details are needed to explain why dopamine monomers (DOPA) demonstrate improved water resistance when combined with butylamine (BA).

As highlighted in a recent study (Cell Reports Physical Science 2023, 4, 101597), the adhesive properties of HPDA are significantly affected by pH levels. The authors should examine the influence of pH on the perovskite precursor.

2. Grain Size: While the observation that HPDA can increase grain size is intriguing, a detailed mechanism is missing. The authors should conduct in-situ or ex-situ analyses using GIWAX, XRD, or PL to investigate how HPDA affects crystallization.

3. Stability of Organic Species: In Figure S15, to substantiate the impact of HPDA on stabilizing organic species within perovskite, TEM images of control samples post-various irradiation durations should be included for comparison.

4. Why is the conductivity of grains in the HPDA-treated film lower than in the control? This raises concerns about whether HPDA might decrease the charge transport capability within perovskite grains. Given that HPDA is an insulator, its effect on the conductivity and mobility within the perovskite film needs to be thoroughly investigated.

5. The mere presence of HPDA on the top and bottom surfaces of the film does not conclusively demonstrate its distribution. Authors should use TOF-SIMS to study the vertical distribution of HPDA within the film.

6. The interaction between HPDA and the contact layers SnOx/Spiro-OMeTAD should be extensively studied through both theoretical and experimental methods.

7. In Figure S30, a detailed explanation is necessary to understand how HPDA influences the energy levels, possibly through theoretical simulations.

8. Given the high PCE reported, it is recommended that the authors certify their device performance for external validation, allowing for direct comparisons with other certified values. The substantial PCE discrepancy (~2%) between the 1-cm² device and smaller devices raises questions about the scalability of this strategy.

9. In Figure 5e, it is important to specify whether the bending tests involve tensile or compressive strain. Additionally, the stability across different bending directions should be compared to assess implications for realistic applications.

10. In Figure S36, the use of rainfall conditions for testing lead capture is questioned since such conditions are not typically considered for FPSCs (Adv. Energy Mater. 2022, 12, 2103236). A more appropriate testing method tailored to FPSCs should be employed.

Reviewer #2

(Remarks to the Author)

In this manuscript, the authors applied a hyperbranched polymer containing various functional groups, including dopamine, to flexible perovskite solar cells (FPSCs). By facilitating specific interactions between the polymer and the perovskite, the performance, moisture resistance, and mechanical stability of the FPSCs were improved. However, several concerns need to be addressed. Major revisions are recommended before considering acceptance.

1. The authors stated that the interaction between SnO₂ and the catechol in the hyperbranched dopamine polymer adhesive (HPDA) is due to hydrogen bonding. Additionally, they mentioned that Spiro-OMeTAD interacts with HPDA via π - π interactions. Clear experimental evidence supporting these chemical interactions should be provided.

2. How about the the molecular weight dependency of HPDA for the performance?

3. In Figure S15, the authors present a spider-web-like structure at the perovskite grain boundary (GB). However, these structures are not easily discernible from the corresponding SEM images. Authors should provide clearer images to support their claims.

4. In Figure S22, the authors demonstrate that photovoltaic parameters vary with the concentration of HPDA in the perovskite precursors. What is the reason for these performance changes?

5. In Figure S30, the energy levels change with the addition of HPDA to the perovskite. Explain the cause of this shift?

6. In Figures 2h and 2i, the thermal conductivity in the GB region of perovskite films containing HPDA is shown to be relatively lower than in the bulk region. However, even in perovskite films without HPDA, the thermal conductivity in the GB region appears low. Authors clarify whether HPDA specifically influences this result.

Reviewer #3

(Remarks to the Author)

Li et al. reports on a hyperbranched polymer additive HPDA added to perovskites. The authors claim that HPDA coordinates with the perovskite grain boundaries and reinforces the perovskite interfaces to improve the fracture toughness and mechanical strength of FPSCs. Hydrophobic HDPDA also contributes to the humidity resistance of the perovskite. The authors show through many characterizations and results the effectiveness of HPDA. The rigid and flexible PSCs attain good PCE, stability, and mechanical durability under a variety of stress tests.

However, the idea is certainly not new. Polymers and additives in general for perovskites have been widely reported in the literature. No novel or surprising results are presented in this work. Actually the same authors have even published a similar paper (Nat. Comm. 14, 6451, 2023) using a different hyperbranched polymer additive to boost the mechanical durability for FPSCs. Perhaps the possible selling point of this work is the improved device results and humidity resistance. But this reviewer is not convinced that just this alone is sufficient to meet the high standards of Nature Communications. Further technical concerns regarding the authors' claims are listed below.

(1) XRD shows that the perovskite crystallinity and preferred orientation is almost identical. Despite the authors' claims, the FWHM is almost unchanged and the small difference might not even be statistically different. Also, why is the background baseline different for fig. S11b? The AFM images in fig. 2h and 2i also show that the grain size is similar between the control and HPDA perovskite. The XRD and AFM results contradict the authors' claims of enhanced crystallinity. More confusingly, only SEM shows that the grain size is larger for the HPDA perovskite, which is also not consistent with the XRD and AFM results. Can the authors explain all these discrepancies?

(2) The authors should provide more evidence of HDPDA distributed along the grain boundaries. The TEM measurement is not convincing. Can the authors perform cross TEM of the perovskite thin film grain boundary? The characterization in fig. 2D is not representative of the perovskite thin film. In fig. 2D, the scraping process would damage the perovskite grains, and the sonication process might redisperse the HDPDA. Is HDPDA soluble in chlorobenzene? Why is the image resolution for the HDPDA perovskite in fig. 2d lower than what the authors showed is capable for the control perovskite in fig. S14. The authors should also provide the corresponding FFT images. Also what is the darker spot in the perovskite region at the bottom of fig. 2D?

(3) The authors claim that HPDA forms a "vertical scaffold from the top to bottom interfaces". Does this mean that the HDPDA forms a continuous vertical distribution from the ETL to HTL interfaces? Can the authors provide tof-sims or xps depth profiling to accurately image the spatial distribution of HPDA? Actually the results show that HTL/perovskite interface Gc increased just slightly from 1.02 to 1.30 with HPDA. Perhaps only a small amount of HPDA is located at the perovskite

surface.

(4) Less grain boundaries lead to better mechanical durability because crack propagation fails through the grain boundaries. This one mechanism could explain most of the observations and results. In this case, how important are the other factors discussed by the author? The authors place a lot of emphasis on the functional groups of HDPAs binding with Pb, SnO₂, and Spiro and the hydrophobicity of the HDPAs chemical structure. But are these important compared to having less grain boundaries? Control experiments with their butylamine based polymer (fig. S6) will help confirm the role of the functional groups of HDPAs.

Minor comments:

(5) The PL peak, UV-Vis bandgap, and EQE cutoff are not consistent. The EQE cutoff shows a redshift of about 20-30nm compared to PL and UV-Vis, which is a large difference.

(6) The authors should provide more information on the in situ PL stability tests.

(7) Does HPDA also work for 1-step perovskites? How about inverted pin devices?

(8) There are some typos in manuscript. For example "spieces relasing" in line 61 and "bondaling" in line 72.

Version 1:

Reviewer comments:

Reviewer #1

(Remarks to the Author)

The revision can be accepted

Reviewer #2

(Remarks to the Author)

Authors properly addressed all the issues raised by this reviewer. Therefore, I recommend this work to be published in the Nature Comm.

Reviewer #3

(Remarks to the Author)

The revised manuscript by Li et al. is much improved. The addition of the GIWAXS, in situ absorption, and HRTEM analysis have clarified the role of HDPAs and its interaction with the perovskite. Generalization of their strategy to the 1-step pin devices has also elevated the impact of this manuscript. Overall, I support the publication of this revised manuscript. I have a few remaining questions:

(1) In the fig. 2D TEM image, what is the bright contrast layer between Spiro and the perovskite surface? I guess it is most likely some passivation layer. Yet, the Materials and Methods section has no mention of a passivation layer.

(2) For the added ToF-SIMS results, how did the authors assign C₆H₄O₂ as the marking indicator for HDPAs? C₆H₄O₂ does not correspond to the chemical formula of dopamine.

(3) Details of the cross section TEM measurement must be clearly written in the Methods section. More generally, the Methods is missing a lot of critical information about the material characterization techniques (e.g. TEM, FIB, GIWAXS, in situ absorption).

Specific reply to Reviewer's comments

Title: **Boosting Mechanical Durability under High Humidity by Bioinspired Multisite Polymer for High-Efficiency Flexible Perovskite Solar Cells**

We sincerely thank all the reviewers for their time, effort and valuable comments. Point-to-point reply to the comment is listed below in blue text and corresponding revision is made in the revised manuscript and provided below in green text.

Reviewer 1:

The authors have incorporated an innovative HPDA with strong adhesion properties to perovskite layers and interfaces, resulting in significant enhancements in mechanical stability under humid conditions. The use of this material group in FPSCs is noteworthy. However, the explanation of the underlying mechanisms is lacking, and certain results require additional validation. The detailed comments are as follows:

Author reply: We thank the reviewer for appreciating the significance of our study and providing valuable comments. We have addressed all comments and believe that the revision has led to an improved manuscript.

1. Mechanism of Underwater Adhesion:

In Figure S5, it is crucial to provide a detailed explanation of the chemical reactions or interactions responsible for the water-triggered coacervation of HPDA adhesives. It is necessary to determine whether coacervation could occur in the perovskite film under ambient conditions and potentially impair charge transport at the grain boundaries.

In Figures 1b and S6, further details are needed to explain why dopamine monomers (DOPA) demonstrate improved water resistance when compared with butylamine (BA).

As highlighted in a recent study (Cell Reports Physical Science 2023, 4, 101597), the adhesive properties of HPDA are significantly affected by pH levels. The authors should examine the influence of pH on the perovskite precursor.

Author reply: We thank the reviewer for this valuable comment. Compared to linear counterparts, hyperbranched polymer has a unique branched 3D structure and denser terminal functional groups. Additionally, hydrophobic interactions can induce spontaneous coacervation in water. Inspired by this, we designed and synthesized a hyperbranched dopamine polymers adhesive (HPDA) with a hydrophobic backbone and adhesive catechol side branches in this work (Fig. S1). As depicted in Fig. R1, when the adhesive is in contact with water, phase transition occurred, which led to coacervation of the hydrophobic interior of HPDA and form coacervates quickly, repelling water molecules out of the backbone of the polymer. At the same time, the contraction of hydrophobic backbone results in increased exposure of the catechol groups outwards, thereby enhancing the interactions of these adhesive moieties with the substrate surface to achieve strong adhesion underwater.

It is shown that the structures of the polymer in solution can be tuned by the solvent (Nat Commun. 2015, 6, 8663; J. Polym. Sci. Part A: Polym. Chem. 2017, 55, 9). The configuration of the HPDA backbone is in stretching state in a good solvent (e.g. DMF). When it contacts with a poor solvent (e.g. water), spontaneous coacervation will take place in the hydrophobic chains of HPDA and form larger aggregates with strong interchain interactions, triggering polymer solidification.

On the other hand, due to the abundant chemical anchor sites of carbonyl (C=O), ether (C-O-C) and hydroxyl (OH-) on the HPDA polymer branches, the polymer form coordination with Pb^{2+} cations in bulk perovskite. Thus, the branches of the HPDA anchor to the perovskite grains and act as a bridge to form a polymer-perovskite composite (Nat. Commun. 2019, 10, 520). When perovskite films are exposed to humid environment, due to the anchoring effect between HPDA and Pb^{2+} , the transformation and aggregation of polymer inside the film is constrained, different from the free transformation in solvent (Chem. Rev., 2010, 110, 1857). Due to the confinement in the perovskite grain boundary, HPDA with 3D dendritic structure will arrange in a way that the catechol head group faces both perovskite, while the middle hydrophobic segment endows the device with overall hydrophobic properties.

To illustrate the roles of dopamine end-groups for achieving the strong adhesion underwater, we prepared another hyperbranched polymers by replacing the dopamine monomer (DOPA) with butylamine (BA), and termed as HPBA (Fig. S6). Hydrophobic interactions of HPBA can induce spontaneous coacervation in water, also causing water repelling from the adhered surface. However, only DOPA-containing coacervates can displace interfacial water molecules from the surface due to its high wetting properties which significantly improve wet-adhesion properties by promoting the formation of molecular hydrogen bonds between catechol groups and SnO_2 surface. The analyses above suggest that an appropriate molecular architecture, hydrophobicity, and distribution density of dopamine may contribute to the best adhesion performance of HPDA under water.

The adhesion performance of the HPDA adhesive was tested across a broad range of pH (Fig. R2a). The HPDA is shown to keep robust adhesion strength over a wide pH range from 3 to 11. In spite of slight decrease in adhesion strength at pH 11 compared with pH 7.4, the HPDA can maintain 196.8 kPa adhesion strengths after soaking for 12 h. As the pH decreases, the adhesion strength is enhanced gradually. At pH 3, the adhesion strengths of the HPDA to the substrate increase up to 251.6 kPa (Fig. R2b). It has been identified that the oxidation state of catechol can affect the adhesion ability of dopamine-based adhesive (Biomaterials 2014, 35, 711; Cell Reports Physical Science 2023, 4101597). Next, the oxidation degree of HPDA in different pH of media was investigated by UV-vis spectroscopy. As shown in Fig. R2c, no oxidation occurs after HPDA solidification at pH 3. Whereas at higher pHs, the degree of oxidation of HPDA increases. This means that phenolic hydroxyl groups are partially oxidized to benzoquinone. However, even with moderate oxidation, the HPDA can still contribute to robust adhesion due to hydrogen bonding interactions in the hyperbranched polymer network with the substrates, thus maintaining a high adhesion strength.

Fig. R1. Schematic illustration of water-triggered strong underwater bioinspired adhesion of hyperbranched polymer and the underlying adhesion mechanism.

Fig. R2. (a) Lap shear curve and (b) Adhesion strength of HPDA bonding to ITO substrates after soaking in different pH media for 12 h. (c) UV-vis absorption spectra of HPDA after solidification in different media with pH 3, 7.4, 9, and 11 for 12 h.

2. Grain Size: While the observation that HPDA can increase grain size is intriguing, a detailed mechanism is missing. The authors should conduct in-situ or ex-situ analyses using GIWAX, XRD, or PL to investigate how HPDA affects crystallization.

Author reply: We thank the reviewer for this valuable comment. Starting from a two-step sequential deposition method, the addition of HPDA in the PbI_2 precursor enables strong coordination between PbI_2 and the function groups of HPDA (Fig. 1d). This interaction weakens the coordination between PbI_2 and DMF/DMSO, leading to larger colloidal in PbI_2 solution (Fig. S9). During the subsequent deposition of PbI_2 , the HPDA polymers support the PbI_2 crystals and avoid their collapsing when solvent evaporated, forming a mesoporous scaffold with penetrating pores. This porous structure facilitates penetration of the ammonium halides solutions and accommodates volume changes of the PbI_2 , promoting complete conversion into high-quality perovskite crystals. Additionally, the robust interaction between the polymer and PbI_2 introduces a higher energy barrier for nucleation, which contributes to the growth of larger perovskite grains.

To investigate the crystallization process of the control and HPDA-modified perovskite films, we conducted in situ UV–vis spectroscopy. As shown in Fig. S11 of the revised manuscript, the absorbance at 700 nm for the HPDA-modified perovskite exhibited a slower increase during the annealing process compared to pristine perovskite. This slower conversion from the intermediate phase to α -FAPbI₃ indicates a strong interaction between HPDA and the perovskite material, delaying the nucleation process.

The use of polymer additives to enhance perovskite grain size has been widely reported, including polymethyl methacrylate (PMMA) (Nat. Energy, 2016, 1, 16142), poly(propylene carbonate) (PPC) (Nat. Commun. 2019, 10, 520), and polarized ferroelectric polymers (Adv. Mater. 2019, 31, 1902222). These mechanisms typically involve two key effects: (1) **Increasing nucleation rate of PbI₂**: The crystal growth from pure PbI₂ solution is more like a homogeneous nucleation process, resulting in a compact PbI₂ film. Adding HPDA in PbI₂ solution forms an intermediate compound with the NH...I hydrogen bonding interactions between HPDA and PbI₂, which facilitate heterogeneous nucleation and lower the nucleation energy of PbI₂, resulting in mesoporous PbI₂ films (Adv. Mater. 2023, 35, 2210186; Adv. Mater. 2022, 34, 2200705). (2) **Retarding perovskite crystallization process**: HPDA could strongly interact with the PbI₂ to form intermediate adducts, and slow down the reaction between organic salt and PbI₂, perovskite grains prefer to grow along the thermodynamically favored directions to minimize Gibbs free energy, which leads to higher crystallinity (Nat. Energy, 2016, 1, 16142). In our work, HPDA plays similar roles promoting heterogeneous nucleation of PbI₂ and guiding oriented growth of perovskite grains, resulting in higher quality perovskite films.

We further analyzed the crystallization properties of perovskite films using grazing incidence wide-angle X-ray scattering (GIWAXS). As shown in the Fig. S13 of revised manuscript, the intensity of the characteristic (110) peak at $q \approx 10 \text{ nm}^{-1}$ increased significantly in HPDA-modified films, demonstrating enhanced crystallinity. Moreover, the PbI₂ peak at $q \approx 9 \text{ nm}^{-1}$ disappeared, which is attributed to complete conversion of PbI₂ to perovskite. To evaluate the final crystal plane orientation, we extracted azimuthal scan profiles for the (110) facets from the GIXRD patterns of the annealed perovskite films. The degree of in-plane orientation was quantitatively assessed by calculating the ratio of oriented crystallites using the formula of $f_{ori} = \frac{A_{ori}}{A_{ori} + A_{iso}}$, where A_{ori} and A_{iso} are the amount of oriented and isotropic crystallites, respectively, obtained from the integrated area of the corresponding fitted Gaussian peaks in Fig. S13d. The f_{ori} of perovskite film increases significantly from 34.2% to 89.6% after HPDA incorporation, indicating that HPDA enables the formation of highly ordered perovskite film with the (110) plane parallel to the substrate.

Fig. S11. In situ absorption spectra of (a) control and (b) HPDA-modified perovskite films. (c) The

in-situ absorbance changes at 700 nm of control and HPDA-modified perovskite films during the thermal annealing process.

Fig. S13. GIXRD patterns of the (a) control and (b) HPDA-modified perovskite films. (c) GIWAXS q -integrated intensity curves for control and HPDA-modified perovskite films. (d) Azimuthal integrated curves of the (110) plane of the perovskite films without and with HPDA.

Revision: In the revised manuscript, we have added the in situ absorption spectra as Fig. S11 in the supporting information and added the description on page 9: “In situ UV–vis spectra of the perovskite films during annealing indicated that the presence of HPDA can form intermediate adducts between organic salt and PbI_2 , guiding oriented growth of perovskite and achieving larger perovskite grains (Fig. S11).”. Also, we have added GIXRD patterns as Fig. S13 in the supporting information, and added the description on page 9: “Moreover, the azimuthal scan profiles for the (110) facets extracted from the GIXRD patterns of the annealed perovskite film indicated HPDA enables the formation of highly ordered perovskite film with the (110) plane parallel to the substrate (Fig. S13).”.

3. Stability of Organic Species: In Figure S15, to substantiate the impact of HPDA on stabilizing organic species within perovskite, TEM images of control samples post-various irradiation durations should be included for comparison.

Author reply: We thank the reviewer for this valuable comment. To address the in-situ cracking of grain boundaries, we have included SEM images of the control samples after various durations of electron beam irradiation. As shown in Fig. S17, obvious cracks appear at the grain boundaries of pure PVK films after irradiation, and the cracking in these films proceeded at a noticeably faster rate.

Fig. S17. Time-dependent high-resolution scanning electron microscope image of control perovskite grain boundaries under intense electron beam irradiation.

Revision: In the revised manuscript, we have added the SEM images of control samples post-various irradiation durations as Fig. S17 in the supporting information. Also, we have added the description on page 10: “However, polymer membrane was not observed at the grain boundary of the control films, and the grain boundaries exhibited a significantly faster cracking rate”.

4. Why is the conductivity of grains in the HPDA-treated film lower than in the control? This raises concerns about whether HPDA might decrease the charge transport capability within perovskite grains. Given that HPDA is an insulator, its effect on the conductivity and mobility within the perovskite film needs to be thoroughly investigated.

Author reply: We thank the reviewer for this valuable comment. It has been well established that the low formation energy of detrimental defects at grain boundaries is a key factor contributing to perovskite degradation. When HPDA is introduced into the perovskite film, the $-\text{COO}$, $-\text{NH}-$, and $-\text{OH}$ functional groups of HPDA effectively passivate grain boundary (GB) defects, as confirmed by our FTIR and XPS results (Fig. 1d, e, and Fig. S8). Furthermore, we have provided supporting evidence through space-charge-limited current (SCLC) measurements (Fig. 3g), dark current leakage analysis (Fig. S36), and trap density of states (tDOS) measurements (Fig. 3h), all of which demonstrate that HPDA effectively passivates defects and reduces trap density.

Additionally, the enhanced photoluminescence peak intensity (Fig. S18) and the prolonged carrier lifetime (Fig. S19) further indicate a significant reduction in defect density and suppression of nonradiative recombination. The TEM image (Fig. 2d) reveals the formation of a polymer-perovskite composite bridge between adjacent grains. This polymer-perovskite interconnection minimizes electrical decoupling or insulation between grains, thereby improving charge transport. As a result, HPDA, functioning as a polymeric Lewis base, enhances the electrical conductivity of the perovskite film, consistent with previous findings (Nat. Commun. 2019, 10, 520).

It is noteworthy that the optimal doping level of polymers in perovskite films is typically very low. For example, prior studies have reported doping levels of 0.02 mg/mL for PU (Adv. Funct. Mater. 2017, 27, 1703061), 0.6 mg/mL for PMMA (Nat. Energy 2016, 1, 16142), 0.2 wt% for P(VDF-TrFE), and 0.3 mg/mL for PVP (Adv. Mater. 2019, 31, 1902222). At such low doping concentrations, the influence of polymer conductivity on device performance is minimal. Jiang et al. (Adv. Energy Mater. 2018, 8, 1701757) introduced trace amounts (0.05%) of various polymers, including representative p-type and n-type semiconducting polymers as well as classical insulators, into perovskite precursor solutions. Their results showed that all polymers effectively passivated defects and enhanced device performance, regardless of their molecular structures and conductivity.

Similarly, Li et al. (Adv. Funct. Mater. 2018, 28, 1706377) demonstrated that both p-type and n-type polymers with different charge transport properties resulted in comparable device performance improvements.

To further investigate the charge transport performance, we conducted conductive atomic force microscopy (C-AFM). As shown in Fig. R3, although the grain boundary exhibited relative lower current density compared to the adjacent grain interior region, the overall HPDA-modified perovskite film exhibited enhanced and homogeneous current signals, suggesting improved charge transport. This enhancement is attributed to reduced trap states and suppressed nonradiative recombination. Additionally, we measured the conductivity of the perovskite layer using interdigital electrodes (Fig. R4). Conductivity was normalized to the electrode geometry and calculated using the following equation:

$$\sigma = \frac{I}{V} \frac{d}{(2n-1)lh}$$

Where I is the measured current, V is the applied voltage, d (0.2 mm) is the spacing between adjacent electrodes, n (6) is the number of finger pairs, l (5 mm) is the length of the overlap area of the fingers, and h (700 nm) is the thickness of the perovskite film. The results (Fig. R4) show that HPDA does not negatively affect perovskite conductivity. On the contrary, the conductivity is slightly improved after HPDA modification, likely due to the polymer-doped perovskite film exhibiting uniform and large grains with reduced trap-state density.

Fig. R3. Conductive atomic force microscope (c-AFM) images of perovskite films with and without HPDA medication.

Fig. R4. Conductivity measurements of perovskite films with and without HPDA modification.

5. The mere presence of HPDA on the top and bottom surfaces of the film does not conclusively demonstrate its distribution. Authors should use TOF-SIMS to study the vertical distribution of HPDA within the film.

Author reply: We thank the reviewer for this valuable comment. In response, we conducted time-of-flight secondary ion mass spectrometry (ToF-SIMS) to analyze the distribution of the HPDA polymer within the perovskite films. Specifically, $C_6H_4O_2^+$ and PbI_2^+ ions were used as indicators for HPDA and perovskite, respectively. As shown in Fig. S21, the distribution of $C_6H_4O_2^+$ confirms that the HPDA polymer is concentrated on both the top and bottom surfaces of the perovskite layer. This finding is consistent with the scanning thermal microscopy results (Fig. 2h, i), AFM phase images (Fig. S20), and AFM force images (Fig. S22). Furthermore, the HPDA polymer exhibits a uniform distribution throughout the perovskite film, extending from the surface to the interior. This observation is corroborated by the TEM image (Fig. S16) and cross-sectional transmission electron microscopy images (Fig. 2d).

Fig. S21. Time of Flight Secondary Ion Mass Spectrometry (ToF-SIMS) depth profiles of the (a) control and (b) HPDA-modified PSCs.

Fig. 2d. Cross-sectional transmission electron microscopy (TEM) images of HPDA-modified PSCs.

Revision: In the revised manuscript, we have added the time of flight secondary-ion mass spectrometry (ToF-SIMS) of control and HPDA-modified perovskite as Fig. S21 in the supporting information, and have added the description on page 11, paragraph 2: “which was also verified by the time-of-flight secondary ions mass spectrometry (ToF-SIMS) (Fig. S21)”. Moreover, we have updated the Fig. 2d in the revised manuscript, and have added the description on page 7: “Cross-sectional high-resolution transmission electron microscopy (TEM) was used to further characterize the distribution of HDPA along the grain boundaries. As shown in **Fig. 2d**, the TEM image showed the HPDA cross-linked perovskite film, which can clearly distinguish the grain boundaries (GBs) between the perovskite grains.”.

6. The interaction between HPDA and the contact layers SnOx/Spiro-OMeTAD should be extensively studied through both theoretical and experimental methods.

Author reply: We thank the reviewer for this valuable comment. Adhesives with high densities of carbonyl and hydroxyl groups are well-documented to enhance adhesive strength, which is

particularly desirable for bonding metals, wood, and engineering plastics. Based on this, we hypothesized that the abundant polar terminal groups in HPDA could form high-density, multiple hydrogen bonds with adjacent surfaces, thereby providing excellent adhesive strength. To investigate the chemical interactions between HPDA and the adjacent layers, we employed X-ray photoelectron spectroscopy (XPS) and Fourier-transform infrared (FTIR) spectroscopy.

Firstly, XPS analysis revealed that, following HPDA modification, the binding energies of the Sn 3d peaks (Fig. S23a) of SnO₂ exhibited a downshift. Additionally, the N 1s peak confirmed the presence of HPDA (Fig. S23b). In the FTIR spectra, the O-Sn-O stretching band at 695 cm⁻¹ shifted to 688 cm⁻¹ after HPDA modification (Fig. S23c). Moreover, the characteristic peaks corresponding to the COO⁻, C-N, and benzene stretching vibrations of HPDA were also observed in the HPDA-SnO₂ composite, and all exhibited significant shifts. These results indicate the formation of strong chemical interactions between HPDA and SnO₂. The improved adhesion strength can be attributed to hydrogen bonding between the polar hydroxyl groups in HPDA and the SnO₂ surface (Adv. Mater. 2022, 34, 2106118).

Secondly, the chemical interaction between HPDA and Spiro-OMeTAD was also examined. π - π interactions have been widely reported in perovskite interface modification (Sci. Adv. 2019, 5, eaav8925; Adv. Funct. Mater. 2022, 32, 2204725). Consistent with these results, we observed π - π interaction XPS peak at binding energy of 291 eV (Fig. S25b). Additionally, the FTIR spectra (Fig. S25c) revealed a slight shift of approximately 15 cm⁻¹ toward a lower wavenumber for the benzene stretching vibration band of HPDA, further supporting the formation of π - π interactions with Spiro-OMeTAD.

These findings suggest that HPDA can form strong chemical interactions with both the SnO₂ and Spiro-OMeTAD layers through hydrogen bonds and π - π interactions. Consequently, HPDA acts as a bridging layer, enhancing the interfaces between the ETL and perovskite, as well as the HTL and perovskite. To further validate the enhanced interfacial adhesion induced by HPDA, we provided quantitative measurements in the manuscript, including AFM force images (Fig. S22), nano-scratch tests (Fig. S24), and double cantilever beam (DCB) tests (Fig. 4c and Fig. S26). These results collectively demonstrate the strong interface bonding facilitated by HPDA.

Fig. S23. X-ray photoelectron spectra (XPS) of (a) Sn 3d and (b) N 1s peaks of SnO₂ and HPDA-modified SnO₂ films. c. Fourier-transform infrared (FTIR) spectra of control and HPDA-modified SnO₂ films.

Fig. S25. X-ray photoelectron spectra (XPS) of C 1s peaks of (a) HPDA and (b) HPDA-doped with Spiro-OMeTAD. c. Fourier-transform infrared (FTIR) spectra of HPDA and HPDA-doped with Spiro-OMeTAD.

Revision: In the revised manuscript, we have added the X-ray photoelectron spectra (XPS) and Fourier-transform infrared (FTIR) spectroscopy as Fig. S23 and Fig. S25 to investigate the interaction between HPDA and the contact layers SnOx/Spiro-OMeTAD in the supporting information. Also, we have added the description on page 13, paragraph 1: “which were confirmed by the XPS and FTIR characterization (Fig. S23).” and page 13, paragraph 1: “which were verified by the XPS and FTIR characterization (Fig. S25)”.

7. In Figure S30, a detailed explanation is necessary to understand how HPDA influences the energy levels, possibly through theoretical simulations.

Author reply: We thank the reviewer for their insightful comments. The observed E_F upshift indicates a more p-type character in the HPDA-modified perovskite film, which can be attributed to the surface defect passivation effect (Science. 2024, 383,1198-1204; Science. 2023, 379,288-294; Nat Energy. 2016, 1, 16142; Phys. Rev. Applied 2014, 2, 034007). The absolute shift in energy levels toward the vacuum level in the HPDA-modified film is likely due to a change in the surface state ratio of lead halide-termination to organic halide-termination. Notably, recent studies have demonstrated that lead iodide-terminated surfaces result in deeper energy levels compared to ammonium iodide-terminated surfaces (Chem. Mater. 2017, 29, 958–968, Nature 2023, 624, 557–563).

PbI₂, with its inappropriate valence band energy (−5.76 eV), is known to cause significant energy level mismatch between perovskite films containing excess unreacted PbI₂ and the hole transport layer (Adv. Mater. 2020, 32, 2000617). In our study, HPDA modification was employed to suppress

the formation of PbI_2 , as shown by the XRD results (Fig. R5a), and to reduce the peak intensity of Pb^0 4f on the perovskite surface (Fig. R5b). Consequently, the perovskite surface is expected to exhibit a higher proportion of organic cation halide termination after HPDA modification and cause upshift of the E_f (Fig. S35).

Fig. R5 (i.e., Supplementary Fig. S12 and Fig. 1e in the original version). (a) XRD patterns of the control and HPDA-modified perovskite films. (b) X-ray photoelectron spectra (XPS) of Pb 4f peaks of control and HPDA-modified perovskite films.

8. Given the high PCE reported, it is recommended that the authors certify their device performance for external validation, allowing for direct comparisons with other certified values. The substantial PCE discrepancy ($\sim 2\%$) between the 1-cm^2 device and smaller devices raises questions about the scalability of this strategy.

Author reply: We thank the reviewer for this valuable comment. We have submitted a certification application to the National PV Industry Measurement and Testing Center (NPVM). However, due to scheduling constraints, it appears that there is currently no available time slot that fully accommodates our requirements. To demonstrate the reliability of the efficiency measurements, the J–V curve was measured using a solar simulator (Newport 94063A Oriel Sol3A, Class AAA) under AM 1.5G (100 mW cm^{-2}) illumination, with light intensity calibrated using a KG-5 filtered silicon standard solar cell (SRC-2020, NREL calibrated). Additionally, to ensure the accuracy of the JSC values obtained from J–V scans, a mask with an aperture area of 0.045 cm^2 was used during the measurement to eliminate any potential overestimation of photocurrent. Furthermore, in our previous work, we provided certified efficiency values for perovskite solar modules (Energy Environ. Sci., 2024, 17, 6302–6313). The results demonstrated minimal deviation between laboratory-tested efficiencies and certified efficiencies, further supporting the accuracy and reliability of our measurement methods.

The up-scaling of the active area in perovskite solar cells (PSCs) is anticipated to result in a decrease in power conversion efficiency (PCE), primarily due to increased series resistance from the electrodes. Initially, our devices were fabricated using ITO substrates with a sheet resistance of $15\ \Omega/\text{sq}$. To enhance device performance, we transitioned to utilizing ITO substrates with a lower sheet resistance of $8\ \Omega/\text{sq}$, which allowed us to achieve a PCE of 25.92%. However, during this phase of optimization, the efficiency improvements for the 1 cm^2 device were not fully valued. To address this oversight, we refabricated the 1 cm^2 devices using the optimized ITO substrate, leading to an improved PCE of 24.68%. Compared to the original device, the most notable enhancement was

observed in the fill factor (FF). To avoid any confusion, we have updated Fig. S30 in the revised supporting information file.

Table R1. Champion photovoltaic parameters of the 1 cm² devices.

	V _{oc} (V)	J _{sc} (mA/cm ²)	FF (%)	PCE (%)
Origin-F	1.168	25.37	78.65	23.32
Origin-R	1.173	25.46	80.36	24.02
Revised-F	1.165	25.58	80.46	23.98
Revised-R	1.176	25.61	81.94	24.68

Fig. S30. J–V curve of the champion control and HPDA-modified PSCs with large area of 1 cm².

9. In Figure 5e, it is important to specify whether the bending tests involve tensile or compressive strain. Additionally, the stability across different bending directions should be compared to assess implications for realistic applications.

Author reply: We thank the reviewer for this valuable comment. Perovskite solar cells (PSCs) are subjected to tensile strain during convex bending. According to the fundamental principles of fracture mechanics (Joule, 2021, 5, 1587–1601), stress (σ_T) across all layers can be expressed as:

$$\sigma_T = \frac{Eh}{2R}$$

where E is Young's modulus of the layer in question, h is the thickness of the f-PSC, and R is the bending radius.

In general, the sign of the bending radius (R) is defined as follows. R is positive as the f-PSC is in convex bending states, generating tensile strain within the f-PSCs. The R is negative as the f-PSCs is in concave bending states, generating compressive stress. Therefore, convex bending of a f-PSC results in tensile stress in all the layers.

Additionally, the mechanical stability of PSCs is influenced by the mode of bending (e.g., concave or convex), as different bending configurations generate distinct stress profiles, which in turn affect the mechanical properties and device performance (J. Phys. Chem. C, 2020, 124, 2340–2345). To further evaluate the improved mechanical stability of HPDA-modified f-PSCs, we conducted fatigue tests under compressive stress by bending the devices in a concave configuration with a curvature radius of 3 mm for 10,000 bending cycles at 65% ambient humidity. Under these conditions, the HPDA-modified f-PSCs retained 96.0% of their original PCE, outperforming the

devices subjected to convex bending (retained 94.1% of their original PCE).

Fig. R6. Normalized PCE for FPSCs as a function of bending cycles by bending in a concave shape (compressive stress) and convex shape (tensile stress) with a bending radius of 3 mm under 65% ambient humidity.

10. In Figure S36, the use of rainfall conditions for testing lead capture is questioned since such conditions are not typically considered for FPSCs (Adv. Energy Mater. 2022, 12, 2103236). A more appropriate testing method tailored to FPSCs should be employed.

Author reply: We thank the reviewer for this valuable comment. We have thoroughly investigated the intrinsic lead-adsorption properties of HPDA and its ability to mitigate lead leakage in perovskite solar cells (PSCs).

Firstly, the lead-adsorption capacity of HPDA was evaluated by adding 2 mL of Pb^{2+} solution (10 ppm) to bottles containing varying amounts of HPDA adsorbent. The bottles were stirred for 30 min to reach equilibrium, and the solution was analyzed using ICP-MS to measure the remaining Pb^{2+} concentration. As shown in Fig. S43a, the lead extraction efficiency increased with the amount of HPDA, reaching 97.5% when 6 mg of HPDA was used.

The lead-adsorption behavior of HPDA films was further quantified through adsorption kinetics measurements. In this test, 1.5×1.5 cm² HPDA/glass samples (net HPDA weight fixed at 6 mg) were immersed face-up in 50.0 mL of PbI_2 aqueous solution (10 ppm) under continuous stirring. Solution samples were collected at specific time intervals (0–360 min) and analyzed by ICP-MS to determine the remaining Pb^{2+} concentration. The amount of Pb adsorbed per unit mass (q_t) of film at the time (t) was determined by the expression $q_t = \frac{V(C_0 - C_t)}{m}$, where C_0 is the initial concentration, C_t is the concentration at time (t), V is the volume of the solution, and m is the mass of the adsorbent.

The adsorption kinetics were analyzed using a pseudo-second-order model: $q_t = \frac{tK_2q_e^2}{1+tK_2q_e}$.

As shown in Fig. S43b, HPDA film adsorbs most of the Pb^{2+} (90%) within 30 min at room temperature. The adsorption kinetics of Pb^{2+} were fitted by a pseudo-second-order model (Figure S43c) based on the data from Fig. S43b. The sorption rate constant (K_2) of HPDA is $5.75 \text{ mg min}^{-1} \text{ g}^{-1}$, and the adsorption capacity at equilibrium (q_e) of HPDA toward Pb^{2+} is 56.49 mg g^{-1} . The high adsorption capacity is attributed to the high density of functional groups in HPDA, while the fast adsorption rate and high capacity suggest that HPDA films can effectively sequester leaked Pb^{2+} from damaged PSCs.

To further evaluate lead leakage prevention, perovskite films were subjected to bending for various cycles, followed by water dripping to measure Pb^{2+} leakage (Fig. 5g). As expected, the Pb^{2+} concentration from the control sample increased with the number of bending cycles due to cracks formed in the perovskite and functional layers. In contrast, HPDA-modified samples showed negligible changes in Pb^{2+} leakage, confirming the capability of HPDA for grain boundary enforcement and lead leakage prevention. Additional tests involved bending flexible PSCs (FPSCs) in different directions (parallel, vertical, and diagonal to the electrode direction) for 500 cycles, followed by water dripping (Fig. S45a). Even under these conditions, HPDA-modified FPSCs maintained low Pb^{2+} concentrations, demonstrating their robustness. To simulate extreme conditions, perovskite films were crumpled 200 times and exposed to water for 30 min. The HPDA-modified films consistently maintained low Pb^{2+} concentrations, highlighting their effectiveness in preventing lead leakage.

Before conducting the lead leakage experiment, we first encapsulated the back and edges of the flexible PSC with 10 μm thick PET tape, and intentionally introduced a hole with diameter of 6 mm on the front side of the FPSC, as shown in Fig. R7. Subsequently, we evaluated lead leakage under acidic conditions by dripping acidic water (pH = 4.2) onto the open hole of FPSCs to simulate acidic rainfall. As shown in Fig. S45b, the Pb^{2+} concentration in contaminated water reached 14.56 ppm for the control sample, while the HPDA-modified FPSCs reduced this to only 0.29 ppm. For further verification, FPSCs subjected to acidic rainfall tests were soaked in 20 mL of acidic water at 85 $^{\circ}\text{C}$ for 3 h to fully dissolve the residual perovskite. As shown in Fig. S45c, Pb^{2+} concentrations from HPDA-modified FPSCs remained low (0.53 ppm), whereas the control sample exhibited a high Pb^{2+} concentration of 82.32 ppm.

These results indicate that HPDA modification prevented 99% of lead leakage under acidic rainfall and thermal stress conditions. Collectively, all tests confirm that HPDA is highly effective in mitigating lead leakage in flexible PSCs, even under extreme mechanical and environmental stress.

Fig. S43. Pb^{2+} absorption properties of HPDA. (a) Influence of absorbent weight toward Pb^{2+} extraction efficiency. (b) Pb^{2+} sorption kinetics and the corresponding absorbed Pb^{2+} amount of HPDA films. (c) The kinetics fitting curves of HPDA films from a pseudo-second-order mode.

Fig. R7. The schematics of flexible PSCs with intentionally introducing a hole with radius of 3mm on the front side of the FPSC for lead leakage test.

Fig. S45. (a) Pb-leakage behavior of the perovskite films after bending with different directions (parallel, vertical, and diagonal to the electrode direction) for 500 cycles and crumpling for 200 times following by water dripping for 30 min, respectively. (b) Acidic water dripping test of control and HPDA-modified perovskite films. (c) Further heat of the control and HPDA-modified samples after (b) acidic rainfall test at 85 °C for 3 h.

Revision: In the revised manuscript, we have added the Pb extraction behavior of HPDA films as Fig. S43 in the supporting information, and have added a sentence on page 21: “due to the high adsorption rate ($5.75 \text{ mg min}^{-1} \text{ g}^{-1}$) and saturated adsorption capacity (56.49 mg g^{-1}) of HPDA (Fig. S43).” Also, we have added Fig. S45 in the supporting information, and have added a sentence on page 21: “The strong lead capturing capability of HPDA was also proven by the results of FPSCs bent with different directions (parallel, vertical, and diagonal to the electrode direction) and crumpled (Fig. S45a). In addition, 99% of lead leakage was prevented by incorporating HPDA modification in the acidic rainfall test and hot water soaking test (Fig. S45b, c). All tests confirm that HPDA can effectively prevent lead leakage in flexible PSCs.”

Reviewer #2 (Remarks to the Author):

In this manuscript, the authors applied a hyperbranched polymer containing various functional groups, including dopamine, to flexible perovskite solar cells (FPSCs). By facilitating specific interactions between the polymer and the perovskite, the performance, moisture resistance, and mechanical stability of the FPSCs were improved. However, several concerns need to be addressed. Major revisions are recommended before considering acceptance.

Author reply: We thank the reviewer for appreciating the significance of our study and providing valuable comments. We have addressed all comments and believe that the revision has led to an improved manuscript.

1. The authors stated that the interaction between SnO₂ and the catechol in the hyperbranched dopamine polymer adhesive (HPDA) is due to hydrogen bonding. Additionally, they mentioned that Spiro-OMeTAD interacts with HPDA via π - π interactions. Clear experimental evidence supporting these chemical interactions should be provided.

Author reply: We thank the reviewer for this valuable comment. Adhesives with high densities of carbonyl and hydroxyl groups are well-documented to enhance adhesive strength, which is particularly desirable for bonding metals, wood, and engineering plastics. Based on this, we hypothesized that the abundant polar terminal groups in HPDA could form high-density, multiple hydrogen bonds with adjacent surfaces, thereby providing excellent adhesive strength. To investigate the chemical interactions between HPDA and the adjacent layers, we employed X-ray photoelectron spectroscopy (XPS) and Fourier-transform infrared (FTIR) spectroscopy.

Firstly, XPS analysis revealed that, following HPDA modification, the binding energies of the Sn 3d peaks (Fig. S23a) of SnO₂ exhibited a downshift. Additionally, the N 1s peak confirmed the presence of HPDA (Fig. S23b). In the FTIR spectra, the O-Sn-O stretching band at 695 cm⁻¹ shifted to 688 cm⁻¹ after HPDA modification (Fig. S23c). Moreover, the characteristic peaks corresponding to the COO⁻, C-N, and benzene stretching vibrations of HPDA were also observed in the HPDA-SnO₂ composite, and all exhibited significant shifts. These results indicate the formation of strong chemical interactions between HPDA and SnO₂. The improved adhesion strength can be attributed to hydrogen bonding between the polar hydroxyl groups in HPDA and the SnO₂ surface (Adv. Mater. 2022, 34, 2106118).

Secondly, the chemical interaction between HPDA and Spiro-OMeTAD was also examined. π - π interactions have been widely reported in perovskite interface modification (Sci. Adv. 2019, 5, eaav8925; Adv. Funct. Mater. 2022, 32, 2204725). Consistent with these results, we observed π - π interaction XPS peak at binding energy of 291 eV (Fig. S25b). Additionally, the FTIR spectrum (Fig. S25c) revealed a slight shift of approximately 15 cm⁻¹ toward a lower wavenumber for the benzene stretching vibration band of HPDA, further supporting the formation of π - π interactions with Spiro-OMeTAD.

These findings suggest that HPDA can form strong chemical interactions with both the SnO₂ and Spiro-OMeTAD layers through hydrogen bonds and π - π interactions. Consequently, HPDA acts as a bridging layer, enhancing the interfaces between the ETL and perovskite, as well as the HTL and

perovskite. To further validate the enhanced interfacial adhesion induced by HPDA, we provided quantitative measurements in the manuscript, including AFM force images (Fig. S22), nano-scratch tests (Fig. S24), and double cantilever beam (DCB) tests (Fig. 4c and Fig. S26). These results collectively demonstrate the strong interface bonding facilitated by HPDA.

Fig. S23. X-ray photoelectron spectra (XPS) of (a) Sn 3d and (b) N 1s peaks of SnO₂ and HPDA-modified SnO₂ films. (c) Fourier-transform infrared (FTIR) spectra of control and HPDA-modified SnO₂ films.

Fig. S25. X-ray photoelectron spectra (XPS) of C 1s peaks of (a) HPDA and (b) HPDA-doped with Spiro-OMeTAD. (c) Fourier-transform infrared (FTIR) spectra of HPDA and HPDA-doped with Spiro-OMeTAD.

Revision: In the revised manuscript, we have added the X-ray photoelectron spectra (XPS) and Fourier-transform infrared (FTIR) spectroscopy as Fig. S23 and Fig. S25 to investigate the interaction between HPDA and the contact layers SnOx/Spiro-OMeTAD in the supporting

information. Also, we have added the description on page 13, paragraph 1: “which were confirmed by the XPS and FTIR characterization (Fig. S23).” and page 13, paragraph 1: “which were verified by the XPS and FTIR characterization (Fig. S25)”.

2. How about the molecular weight dependency of HPDA for the performance?

Author reply: We synthesized three HPDA polymers with different molecular weights by adjusting the monomer ratio and polymerization time. As shown in Fig. R8, the molecular weights of the synthesized HPDA polymers are 5,217 Da, 9972 Da and 17,780 Da, respectively. Polymer with molecular weight of 9972 Da was used in the original manuscript.

We investigated the influence of HPDA molecular weight on the performance of flexible perovskite solar cells (FPSCs). As shown in Fig. R8d, the molecular weight of HPDA has a significant influence on the power conversion efficiency (PCE) of FPSCs. A moderate increase in HPDA molecular weight from 5217 to 9972 leads to a notable enhancement in PCE. This improvement can be attributed to the longer HPDA chains, which facilitate the formation of a three-dimensional molecular network within the perovskite film, ensuring uniform morphology and improved film quality. However, a further increase in molecular weight beyond 9972 introduces a stronger steric hindrance, which disrupts charge transport, thereby resulting in decreased PCE. Consequently, PSCs based on HPDA with a medium molecular weight exhibited the best device performance.

These findings align with previous studies reporting that the molecular weight of polymer additives plays a crucial role in perovskite film quality and device performance. For instance, Zhao et al. (Nat. Commun. 2016, 7, 10228) demonstrated that the optimal molecular weight of PEG dopants was above 20,000 Da, as lower molecular weights (e.g., 12,000 Da) not only failed to enhance performance but also negatively affected the device. Similarly, Li et al. (ACS Appl. Energy Mater. 2022, 5, 12158–12164) found that perovskite grain size increased with the molecular weight of PEG additives, which significantly reduced dark current and noise current. However, an excessively high molecular weight resulted in a decrease in grain size and an increase in trap density due to disrupted crystallization and steric effects.

In our case, the medium molecular weight of HPDA (9972 Da) achieves an optimal balance between forming a robust three-dimensional molecular network and avoiding excessive steric hindrance, leading to the highest PCE among the tested devices. This observation is consistent with the literature and highlights the importance of tuning polymer molecular weight to optimize perovskite film morphology and device performance.

Fig. R8. Gel permeation chromatography (GPC) curves of HPDA with different molecular weights (a) low, (b) medium and (c) high. (d) J-V curves of the reverse scan based on precursor solution with different HPDA molecular weight.

3. In Figure S15, the authors present a spider-web-like structure at the perovskite grain boundary (GB). However, these structures are not easily discernible from the corresponding SEM images. Authors should provide clearer images to support their claims.

Author reply: We thank the reviewer for this valuable comment. To address the concern, we have provided clearer images of the perovskite film to illustrate the spider-web-like structure at the perovskite grain boundaries (GBs).

Additionally, we have included cross-sectional high-resolution transmission electron microscopy (HR-TEM) images to further identify the specific location of the HPDA polymer (Fig. S17). In the low-magnification TEM image, the HPDA-crosslinked perovskite film is clearly visible, and the grain boundaries between perovskite grains can be distinctly identified. To better understand the distribution of HPDA, we analyzed two representative regions near the GBs. The results reveal the presence of an ultra-thin amorphous layer at the GBs, which corresponds to the HPDA polymer. This polymer layer serves as a bridge between adjacent perovskite grains, promoting grain connectivity and enhancing interfacial adhesion.

Furthermore, high-resolution TEM analysis shows the perovskite lattice fringes with an interplanar spacing of 0.31 nm, corresponding to the (110) plane of the tetragonal phase of the MAFA crystal. This indicates that the long-chain HPDA polymer interacts with the perovskite material to form a polymer-perovskite composite interconnection structure. These results demonstrate that HPDA preferentially localizes at the grain boundaries of the perovskite film, where it plays a crucial role in bridging adjacent grains and reinforcing the perovskite structure.

Fig. S17. Time-dependent high-resolution scanning electron microscope image of (a) control and (b) HPDA-modified perovskite grain boundaries under intense electron beam irradiation.

Fig. 2d. Cross-sectional transmission electron microscopy (TEM) images of HPDA-modified PSCs.

Revision: We have added the SEM images of control perovskite post-various irradiation durations and updated the SEM images of HPDA-modified perovskite as Fig. S17 in the revised supporting information file. Also, we have added the description in the revised manuscript (page 10): “As shown in Fig. S17, with the prolongation of the exposure time, perovskite films with HPDA showed a polymer network membrane inside the GBs, which tightly bind the grains together. However, polymer network membrane was not observed within the control films, and the grain boundaries exhibited a significantly faster cracking rate.”

4. In Figure S22, the authors demonstrate that photovoltaic parameters vary with the concentration of HPDA in the perovskite precursors. What is the reason for these performance changes?

Author reply: The introduction of polymers improves the quality of perovskite films due to their strong passivation effects. However, the doping amount of the polymer is critical for achieving optimal device performance. Insufficient polymer content results in weak passivation, while excessive polymer loading can cause poor charge transfer and increased series resistance. When the doping amount exceeds the optimal level, phase separation between the polymer and perovskite domains occurs, which can lead to polymer precipitation. This results in the formation of pinholes and polymer clusters on the film surface, ultimately deteriorating device performance.

Additionally, the polymer doping amount directly affects the viscosity of the precursor solution, which significantly influences the thickness and uniformity of the perovskite film. The addition of polymer also impacts the crystallinity and morphology of the PbI_2 film, promotes the reaction

between PbI_2 and FAI, and alters the crystallization kinetics of the perovskite. However, excessive polymer content hinders perovskite crystal growth because long polymer chains cannot freely "migrate" within the film, resulting in incomplete PbI_2 conversion.

Furthermore, device performance variations are closely linked to changes in trap states within the perovskite film, which are influenced by the polymer doping amount. This is supported by the photoluminescence (PL) intensity measurements (Fig. R9), which reveal a strong correlation between polymer content and the density of trap states across the perovskite film.

Fig. R9. Steady-state PL spectra of perovskite films with different modification concentration (mg ml^{-1}) of HPDA

5. In Figure S30, the energy levels change with the addition of HPDA to the perovskite. Explain the cause of this shift?

Author reply: We thank the reviewer for their insightful comments. The observed EF upshift indicates a more p-type character in the HPDA-modified perovskite film, which can be attributed to the surface defect passivation effect (Science. 2024, 383,1198-1204; Science. 2023, 379,288-294; Nat Energy. 2016, 1, 16142; Phys. Rev. Applied 2014, 2, 034007). The absolute shift in energy levels toward the vacuum level in the HPDA-modified film is likely due to a change in the surface state ratio of lead halide-termination to organic halide-termination. Notably, recent studies have demonstrated that lead iodide-terminated surfaces result in deeper energy levels compared to ammonium iodide-terminated surfaces (Chem. Mater. 2017, 29, 958–968, Nature 2023, 624, 557–563).

PbI_2 , with its inappropriate valence band energy (-5.76 eV), is known to cause significant energy level mismatch between perovskite films containing excess unreacted PbI_2 and the hole transport layer (Adv. Mater. 2020, 32, 2000617). In our study, HPDA modification was employed to suppress the formation of PbI_2 , as shown by the XRD results (Fig. R5a), and to reduce the peak intensity of Pb^0 4f on the perovskite surface (Fig. R5b). Consequently, the perovskite surface is expected to exhibit a higher proportion of organic cation halide termination after HPDA modification and cause upshift of the E_f (Fig. S35).

Fig. R5 (i.e., **Supplementary Fig. S12** and **Fig. 1e** in the original version). (a) XRD patterns of the control and HPDA-modified perovskite films. (b) X-ray photoelectron spectra (XPS) of Pb 4f peaks of control and HPDA-modified perovskite films.

6. In Figures 2h and 2i, the thermal conductivity in the GB region of perovskite films containing HPDA is shown to be relatively lower than in the bulk region. However, even in perovskite films without HPDA, the thermal conductivity in the GB region appears low. Authors clarify whether HPDA specifically influences this result.

Author reply: The thermal conductivity line profile along the grain boundaries are recorded from the black solid line in the corresponding SThM thermal images. As shown in Fig. R10, control perovskite exhibits a difference ($\Delta\lambda \approx 13$) between the grain boundaries (GBs) and adjacent grain interior. The GBs show a low thermal conductivity, because the GBs are highly disordered with large number of defects (Nat Energy. 2020, 7, 794; Science. 2020, 370, eabb5940). The HPDA-modified perovskite exhibits a larger difference ($\Delta\lambda \approx 34$) between the GBs and adjacent grain interior, in which the local GBs show a relatively lower thermal conductivity than control perovskite. In most of the cases, polymers presented lower bulk thermal conductivity (Adv. Funct. Mater. 2020, 30, 1900892). Therefore, this evidences clearly indicate that HPDA is assembled at GBs to further decrease the thermal conductivity of GBs.

Fig. R10. (i.e., Fig. 2h, i in the original version). SThM thermal images and thermal conductivity curves vary with distance recorded from the black solid line in the corresponding SThM thermal images of (a) control and (b) HPDA-modified perovskite films on the top and bottom surfaces

Reviewer #3 (Remarks to the Author):

Li et al. reports on a hyperbranched polymer additive HPDA added to perovskites. The authors claim that HPDA coordinates with the perovskite grain boundaries and reinforces the perovskite interfaces to improve the fracture toughness and mechanical strength of FPSCs. Hydrophobic HDPA also contributes to the humidity resistance of the perovskite. The authors show through many characterizations and results the effectiveness of HPDA. The rigid and flexible PSCs attain good PCE, stability, and mechanical durability under a variety of stress tests.

However, the idea is certainly not new. Polymers and additives in general for perovskites have been widely reported in the literature. No novel or surprising results are presented in this work. Actually the same authors have even published a similar paper (Nat. Comm. 14, 6451, 2023) using a different hyperbranched polymer additive to boost the mechanical durability for FPSCs. Perhaps the possible selling point of this work is the improved device results and humidity resistance. But this reviewer is not convinced that just this alone is sufficient to meet the high standards of Nature Communications. Further technical concerns regarding the authors' claims are listed below.

Thank you for your comment, and we fully appreciate your concern in the novelty of our work over the literature. As is also pointed out by the Referee, polymer-based additives can serve as adhesives to glue the grain boundary, such as polymethyl methacrylate (PMMA) (Nat. Energy, 2016,1,16142), poly(propylene carbonate) (PPC) (Nat. Commun. 2019,10, 520) and polyvinylpyrrolidone (PVP) (Science Advances. 2017, 3, e1700106), which can stitch cracks at the GB regions, thus improving the mechanical robustness of FPSCs. While polymer adhesives have been introduced to enhance mechanical robustness, most polymer adhesives need a dry surface and environment to demonstrate the strongest adhesion ability, the solution processed perovskite films inevitably trap polar solvent molecules in the grain boundary and charge transport layer interfaces of the perovskite films, resulting in the polymer adhesives fail to perform adequately in the polar solvent environments used during perovskite film fabrication. Even worst, the hygroscopic nature of perovskite materials can attract moisture from the ambient air and weaken the adhesion strength of additives over time.

Unlike our previous work that emphasized interface adhesion and stress release for improving mechanical stability, our current work was inspired by marine mussels can firmly adhere to surfaces in seawater via dopamine, and aim to underscores the pivotal role of establishing an interface and grain boundary adhesive layer under high humidity to impede accelerated degradation caused by the cooperative effects of water and mechanical stress, thereby enhancing the efficiency and mechanical stability of FPSCs. This study provides valuable insights and guidelines for designing FPSCs that meet both mechanical and electrical performance requirements, therefore, we believe our work is of high novelty, beyond the scope of the published works.

(1) XRD shows that the perovskite crystallinity and preferred orientation is almost identical. Despite the authors' claims, the FWHM is almost unchanged and the small difference might not even be statistically different. Also, why is the background baseline different for fig. S11b? The AFM images in fig. 2h and 2i also show that the grain size is similar between the control and HPDA perovskite. The XRD and AFM results contradict the authors' claims of enhanced crystallinity. More confusingly, only SEM shows that the grain size is larger for the HPDA perovskite, which is also not consistent with the XRD and AFM results. Can the authors explain all these discrepancies?

Author reply: We thank the reviewer for this valuable comment. The increased intensity ratio of the (110) and (202) XRD peaks (Fig. S12) demonstrates the preferential orientation of perovskite crystals along the vertical direction. This preferential orientation is facilitated by the 3D HPDA scaffold, which acts as a template for crystal growth. When comparing the full width at half maximum (FWHM) of the XRD peaks, we initially did not subtract the background baseline during the normalization process, leading to inconsistencies in the baseline levels. To address this, we have subtracted the baseline and updated Fig. S12b in the revised supporting information file (page 12) to ensure clarity and consistency.

To further investigate the crystallization properties of the perovskite films, we conducted grazing incidence wide-angle X-ray scattering (GIWAXS) measurements. As shown in Fig. S13, the intensity of the characteristic (110) diffraction peak at $q \approx 10 \text{ nm}^{-1}$ increased significantly for the HPDA-modified films, indicating enhanced crystallinity. Additionally, the disappearance of the PbI_2 peak at $q \approx 9 \text{ nm}^{-1}$ is attributed to the strong interaction between HPDA and Pb^{2+} , which effectively passivates grain boundary defects and results in higher film quality.

To evaluate the final crystal plane orientation, we extracted azimuthal scan profiles for the (110) facets from the GIXRD patterns of the annealed perovskite films. The degree of in-plane orientation was quantitatively assessed by calculating the ratio of oriented crystallites using the formula of $f_{ori} = \frac{A_{ori}}{A_{ori} + A_{iso}}$, where A_{ori} and A_{iso} are the amount of oriented and isotropic crystallites, respectively, obtained from the integrated area of the corresponding fitted Gaussian peaks in Fig. S13d. The f_{ori} of perovskite film increases significantly from 34.2% to 89.6% after HPDA incorporation, indicating that HPDA enables the formation of highly ordered perovskite film with the (110) plane parallel to the substrate.

Regarding the scanning electron microscopy (SEM) measurements, the perovskite films had been stored in ambient conditions for several days before characterization, which led to partial decomposition and the appearance of a larger number of bright crystals. To address this, we performed SEM measurements on a freshly prepared perovskite film, as shown in Fig. 2a. Additionally, we conducted in-situ absorption spectroscopy (Fig. S11) to study the crystallization process. Compared with pristine perovskite, the absorbance of the HPDA-modified perovskite film exhibited a slower increase during the annealing process. This delayed conversion from the intermediate phase to $\alpha\text{-FAPbI}_3$ is attributed to the strong interaction between HPDA and the perovskite materials, which slows the crystallization process and results in larger grain sizes.

To further quantify the grain size, we analyzed atomic force microscopy (AFM) images. The average grain size (D) of the control perovskite film was approximately 610 nm, whereas the HPDA-modified perovskite film exhibited a significantly larger grain size of approximately 882 nm (Fig. R10). This observation is consistent with the results obtained from XRD, GIWAXS, and SEM analyses.

Fig. S12. Crystal structure of the control and HPDA-modified perovskite films: (a) XRD patterns of perovskite films; (b) the full width at half maximum (FWHM) of the (110) peak.

Fig. S13. GIXRD patterns of the (a) control and (b) HPDA-modified perovskite films. (c) GIWAXS q-integrated intensity curves for control and HPDA-modified perovskite films. (d) Azimuthal integrated curves of the (110) plane of the perovskite films without and with HPDA.

Fig. S11. In situ absorption spectra of (a) control and (b) HPDA-modified perovskite films. (c) The in-situ absorbance changes at 700 nm of control and HPDA-modified perovskite films during the thermal annealing process.

Fig. R10. (i.e., Fig. S15 in the original version). AFM images of (a) control and (b) HPDA-modified perovskite films. The grain size statistical distribution of the (c) control and (d) HPDA-modified perovskite in AFM images.

Revision: In the revised manuscript, we have updated the Fig. 2a and Fig. S12 in the supporting information accordingly. Also, we add the Fig. S11 and Fig. S13 on page 12 of the revised supporting information file.

(2) The authors should provide more evidence of HPDA distributed along the grain boundaries. The TEM measurement is not convincing. Can the authors perform cross TEM of the perovskite thin film grain boundary? The characterization in fig. 2D is not representative of the perovskite thin film. In fig. 2D, the scraping process would damage the perovskite grains, and the sonication process might redisperse the HPDA. Is HPDA soluble in chlorobenzene? Why is the image resolution for the HPDA perovskite in fig. 2d lower than what the authors showed is capable for the control perovskite in fig. S14. The authors should also provide the corresponding FFT images. Also what is the darker spot in the perovskite region at the bottom of fig. 2D?

Author reply: We thank the reviewer for this valuable comment. As suggested, we have performed cross-sectional high-resolution transmission electron microscopy (TEM) to characterize the distribution of HPDA along the grain boundaries (GBs). As shown in Fig. S16, the low-magnification TEM image reveals the HPDA cross-linked perovskite film, with clearly distinguishable grain boundaries between adjacent perovskite grains. Detailed analysis of two representative regions around the GBs reveals the presence of an ultra-thin amorphous layer, corresponding to the HPDA polymer layer. This layer acts as a bridge between adjacent perovskite grains, facilitating improved connectivity.

High-resolution TEM further confirms the perovskite lattice fringes, with an interplanar spacing of 0.31 nm, corresponding to the (110) plane of the tetragonal phase of the MAFA crystal. These findings indicate that the long-chain HPDA polymer interacts with the perovskite material to form a polymer-perovskite composite interconnection structure. This structure passivates the perovskite

grain boundaries effectively, contributing to enhanced device performance.

Regarding our previous method of TEM samples preparation, the scraping method has been widely reported in the literature (Nat. Commun. 2019, 10, 520; Energy Environ. Sci. 2021, 14, 5406–5415). The crystalline phase of the perovskite is clearly observed, indicating that the scraping process does not damage the perovskite grains. An amorphous region is also distinguishable between contiguous grains, suggesting the accumulation of polymer at the grain boundaries. This observation aligns with previous reports on polymer additives (Nat. Commun. 2018, 9, 3806), further validating the passivation effect of HPDA at the grain boundaries.

Additionally, the HPDA polymer demonstrates extremely low solubility in chlorobenzene (CB), making it insoluble during dispersion. This property ensures that HPDA molecules adhere firmly to the perovskite grains. As a result, the sonication process does not cause redistribution or redispersion of HPDA.

Regarding the image in Fig. R11 (Fig. 2d in the original version), the original image was cropped and compressed to meet size requirements, which caused a slight reduction in resolution. To address this, we have now included the original unmodified image and the corresponding FFT images in the revised manuscript. The darker spot observed in the perovskite region at the bottom corresponds to stacked perovskite grains. As this region is too thick, it appears as a darker area in the TEM image.

Fig. 2d. Cross-sectional transmission electron microscopy (TEM) images of HPDA-modified PSCs.

Fig. R11. TEM images of the HPDA-modified perovskite grain boundary and corresponding FFT images.

Revision: In the revised manuscript, we have updated the Fig. 2d and have added a sentence: “Cross-sectional high-resolution transmission electron microscopy (TEM) was used to further characterize the distribution of HPDA along the grain boundaries. As shown in Fig. 2d, the TEM image showed the HPDA cross-linked perovskite film, which can clearly distinguish the grain boundaries (GBs) between the perovskite grains.” on page 10 of the revised manuscript. Also, we have included the original unmodified image as Fig. S16 in the revised supporting files.

(3) The authors claim that HPDA forms a “vertical scaffold from the top to bottom interfaces”. Does this mean that the HPDA forms a continuous vertical distribution from the ETL to HTL interfaces? Can the authors provide tof-sims or xps depth profiling to accurately image the spatial distribution of HPDA? Actually the results show that HTL/perovskite interface Gc increased just slightly from 1.02 to 1.30 with HPDA. Perhaps only a small amount of HPDA is located at the perovskite surface.

Author reply: We thank the reviewer for this valuable comment. To analyze the distribution of HPDA polymer in perovskite films, we utilized time-of-flight secondary-ion mass spectrometry (ToF-SIMS). Specifically, $C_6H_4O_2^+$ and PbI_2^+ ions were used as indicators for HPDA and perovskite, respectively. As shown in Fig. S21, the distribution of $C_6H_4O_2^+$ confirms that the HPDA polymer is localized on both the top and bottom surfaces of the perovskite layer. This finding is consistent with the scanning thermal microscopy results (Fig. 2h, i), AFM phase images (Fig. S20), and AFM force images (Fig. S22). Furthermore, the HPDA polymer exhibits a uniform distribution throughout the perovskite film, extending from the surface to the interior. This observation is corroborated by the TEM image (Fig. S16) and cross-sectional transmission electron microscopy images (Fig. 2d)

To investigate the chemical interactions between HPDA and the adjacent layers, we employed X-ray photoelectron spectroscopy (XPS) and Fourier-transform infrared (FTIR) spectroscopy. Firstly, XPS analysis revealed that, following HPDA modification, the binding energies of the Sn 3d peaks (Fig. S23a) of SnO_2 exhibited a downshift. Additionally, the N 1s peak confirmed the presence of HPDA (Fig. S23b). In the FTIR spectra, the O-Sn-O stretching band at 695 cm^{-1} shifted to 688 cm^{-1} after HPDA modification (Fig. S23c). Moreover, the characteristic peaks corresponding to the COO^- , C-N, and benzene stretching vibrations of HPDA were also observed in the HPDA- SnO_2 composite, and all exhibited significant shifts. These results indicate the formation of strong chemical interactions between HPDA and SnO_2 . The improved adhesion strength can be attributed to hydrogen bonding between the polar hydroxyl groups in HPDA and the SnO_2 surface (Adv. Mater. 2022, 34, 2106118).

Secondly, the chemical interaction between HPDA and Spiro-OMeTAD was also examined. π - π interactions have been widely reported in perovskite interface modification (Sci. Adv. 2019, 5, eaav8925; Adv. Funct. Mater. 2022, 32, 2204725). Consistent with these results, we observed π - π interaction XPS peak at binding energy of 291 eV (Fig. S25b). Additionally, the FTIR spectra (Fig. S25c) revealed a slight shift of approximately 15 cm^{-1} toward a lower wavenumber for the benzene stretching vibration band of HPDA, further supporting the formation of π - π interactions with Spiro-OMeTAD.

Therefore, our findings demonstrate that HPDA forms strong chemical interactions with both SnO_2 and Spiro-OMeTAD layers via hydrogen bonds and coordination bonds, effectively bridging the ETL and HTL with the perovskite layer and strengthening the interfaces. Furthermore, the energy associated with hydrogen bonds ranges from approximately 10–40 kJ/mol, whereas π - π stacking effects are typically weaker, with energy values lower than 10 kJ/mol (J. Chem. Sci. 2016, 128, 1571–1577). Consequently, the enhancement effect at the ETL/perovskite interface is stronger than at the HTL/perovskite interface, further supporting the superior performance of HPDA at the ETL/PVK interface.

Fig. S21. Time of Flight Secondary Ion Mass Spectrometry (ToF-SMIS) depth profiles of the (a) control and (b) HPDA-modified PSCs.

Fig. 2d. Cross-sectional transmission electron microscopy (TEM) images of HPDA-modified PSCs.

Fig. S23. X-ray photoelectron spectra (XPS) of (a) Sn 3d and (b) N 1s peaks of SnO₂ and HPDA-modified SnO₂ films. c. Fourier-transform infrared (FTIR) spectra of control and HPDA-modified SnO₂ films.

Fig. S25. X-ray photoelectron spectra (XPS) of C 1s peaks of (a) HPDA and (b) HPDA-doped with Spiro-OMeTAD. (c) Fourier-transform infrared (FTIR) spectra of HPDA and HPDA-doped with Spiro-OMeTAD.

(4) Less grain boundaries lead to better mechanical durability because crack propagation fails through the grain boundaries. This one mechanism could explain most of the observations and results. In this case, how important are the other factors discussed by the author? The authors place a lot of emphasis on the functional groups of HDPA binding with Pb, SnO₂, and spiro and the hydrophobicity of the HDPA chemical structure. But are these important compared to having less grain boundaries? Control experiments with their butylamine based polymer (fig. S6) will help confirm the role of the functional groups of HDPA.

Author reply: We thank the reviewer for this valuable comment. As reported in the literature, the fracture energy (G_c) of perovskite films increases with grain size, and coarse-grained perovskite films are generally considered desirable for improving mechanical reliability (Adv. Energy Mater. 2018, 8, 1702116; Extreme Mech. Lett. 2016, 9, 353–358). However, perovskite films prepared using low-temperature solution processes contain internal defects at the grain boundaries (GBs). Under mechanical deformation such as bending, stretching, or twisting, GBs act as stress concentration regions (Nat. Energy 2022, 7, 794). The fracture resistance of perovskite films is a function of the density and distribution of defects and GBs, as defect-laden boundaries facilitate crack propagation with minimal resistance, resulting in reduced cohesion (Academic Press Inc. 1977, 11, 199–381). GBs are also more susceptible to moisture ingress and crack formation, which significantly compromises mechanical stability (Int. J. Plasticity 2021, 27, 801).

The influence of grain boundaries and size on perovskite cohesion suggests that reinforcing these weak boundaries by cross-linking crystalline domains is a promising strategy for increasing fracture resistance (Nat. Chem. 2015, 7, 703–711). Furthermore, cracks and stresses induced by deformation create accelerated pathways for the diffusion of water and oxygen, leading to irreversible perovskite degradation through processes such as organic cation deprotonation and the formation of HI. The volatile by-products and decomposed molecules can build up internal pressure, accelerating

delamination at interfaces and reducing mechanical stability. In particular, weak interfaces in multilayered perovskite solar cells (PSCs) act as degradation pathways for water molecules, making interface adhesion and perovskite hydrophobicity critical to improving the mechanical reliability of flexible PSCs (FPSCs).

Enhanced Mechanical Stability with HPDA:

To address these challenges, we incorporated hyperbranched dopamine-based polymers (HPDA) into the perovskite films to enhance their hydrophobicity and reinforce their mechanical stability under humid conditions. Strong interface adhesion, facilitated by HPDA, is essential for maintaining the efficiency and reliability of PSCs under mechanical and environmental stress.

To investigate the impact of grain size on mechanical properties, we fabricated large-grained perovskite films by introducing MACl. As shown in Fig. R12, SEM images reveal significant morphological changes in perovskite films exposed to humid environments, which adversely affect their mechanical properties. This result highlights the critical role of HPDA's hydrophobic chemical structure in preserving the film's integrity.

We further evaluated the mechanical robustness of MACl-doped FPSCs through fatigue tests conducted under a curvature radius of 3 mm and 10,000 bending cycles at 65% relative humidity. As shown in Fig. R13, the MACl-doped flexible PSCs retained 51.9% of their original PCE under the same conditions, emphasizing the importance of HPDA's hydrophobic structure and its role in improving bending resistance.

Roles of dopamine groups:

In addition, we incorporated a butylamine-based polymer (HPBA) into the perovskite films to further evaluate the role of the dopamine groups in HPDA in enhancing mechanical stability. The oxygen-containing functional groups in HPOA coordinated with Pb^{2+} , forming cross-linked grain boundaries and improving resistance to humidity-induced degradation. Fatigue tests on HPBA-modified FPSCs under identical conditions (curvature radius of 3 mm, 10,000 bending cycles, and 65% ambient humidity) demonstrated enhanced mechanical robustness, with 87.2% of the original PCE retained (Fig. R13).

However, the HPDA-modified FPSCs outperformed HPBA-modified devices, retaining 94.1% of their original PCE. The superior performance of HPDA can be attributed to its unique chemical structure. The dopamine end groups in HPDA enable bidentate hydrogen bonding with SnO_2 , while the benzene ring in the DOPA structure facilitates π - π interactions with Spiro-OMeTAD, resulting in strong adhesion at the ETL/PVK and HTL/PVK interfaces. Additionally, the dopamine end groups allow HPDA to maintain strong adhesion even in humid environments, inhibiting accelerated degradation caused by the combined effects of water ingress and mechanical stress.

Conclusion:

The enhanced bending resistance of HPDA-modified FPSCs under humid conditions originates from the overall improved toughness and hydrophobicity of the perovskite film. By reinforcing grain boundaries, increasing interface adhesion, and providing water resistance, HPDA plays a critical role in improving the mechanical stability and reliability of flexible PSCs, especially under harsh environmental conditions. These findings emphasize the potential of HPDA-based strategies for advancing the durability and performance of perovskite solar cells in practical applications.

Fig. R12. SEM images of (a) MACl-doped and (b) HPDA-doped perovskite film after exposed to 85% humidity for different time.

Fig. R13. Normalized PCE for MACl-doped, HPBA-doped and HPDA-doped FPSCs as a function of bending cycles with a bending radius of 3 mm under 65% relative humidity.

Minor comments:

(5) The PL peak, UV-Vis bandgap, and EQE cutoff are not consistent. The EQE cutoff shows a redshift of about 20-30nm compared to PL and UV-Vis, which is a large difference.

Author reply: We thank the reviewer for this valuable comment. For the initial film characterization, we use the $(\text{FAPbI}_3)_{1-x}(\text{MAPbBr}_3)_x$ perovskite precursor solution with 1.5 M PbI_2 and FAI: MABr: MACl (90:9:9 mg in 1 mL IPA) solution. The corresponding perovskite film exhibited a bandgap of 1.56 eV. However, for the champion cell fabrication, we changed the ammonium salt solution in the second step to FAI: MACl (90:18 mg in 1 mL IPA) solution. This adjustment resulted in a slightly lower bandgap of 1.52 eV for the final perovskite film. The difference in ammonium salt composition between the initial film and the champion device accounts for the discrepancy between the photoluminescence (PL) and UV-Vis bandgap values obtained from the initial perovskite recipe and the external quantum efficiency (EQE) results from the champion cell. To clarify this point, we have re-tested the PL spectra and UV-Vis absorption spectra for the champion perovskite films. As shown in Fig. S14 and Fig. S18, the re-measured UV-Vis and PL spectra demonstrate no significant differences or shifts in bandgap (Fig. R14). To avoid any confusion, we have updated Fig. S14 and Fig. S18 in the revised supporting information file.

Fig. S14. UV-Vis absorption spectra of the perovskite films with and without HPDA.

Fig. S18. Steady-state PL spectra of perovskite films with and without HPDA.

Fig. R14. Comparison of bandgap of perovskite films with and without HPDA.

Revision: In the revised manuscript, we have updated the Fig. S14 and Fig. S18 in the supporting information accordingly.

(6) The authors should provide more information on the in situ PL stability tests.

Author reply and revision: We thank the reviewer for this valuable comment. In the revised manuscript, we have added the in situ PL stability test “The bending durability of f-PSCs was evaluated using a mechanical tester (PR-BDM-100, Puri, China) in constant-radius bending mode with a bending radius of 3 mm in an 85% humidity environment. A homemade system equipped with a fiber optic spectrometer (Ocean Optics, QE Pro) was positioned above the perovskite sample to record the PL spectra at various bending intervals. The corresponding PL spectra were

periodically measured in the flat state to ensure consistent positioning during the test.” on page 4 in method section of the revised supporting information file.

This addition provides a comprehensive description of the in situ PL stability testing process and ensures clarity for the reviewer and readers.

(7) Does HPDA also work for 1-step perovskites? How about inverted pin devices?

Author reply: For the preparation of 1-step inverted p-i-n devices, $\text{Cs}_{0.05}(\text{FA}_{0.95}\text{MA}_{0.05})_{0.95}\text{Pb}(\text{I}_{0.95}\text{Br}_{0.05})_3$ perovskite films were prepared following the previous report (Adv. Mater., 2024, 2407433). Perovskite precursor solutions were prepared by dissolving FAI (232.95 mg), MABr (8.13 mg), PbI_2 (677.07 mg), CsI (19.98 mg), PbBr_2 (28.07 mg), and MACl (15 mg) in 1 mL of a DMF/DMSO (4/1) mixed solvent. For HPDA modification (0.01 mg/ml), HPDA was directly added into perovskite solution.

The architecture of the inverted device is ITO/SAM/Perovskite/ C_{60} /BCP/Ag. The ITO substrates were treated with UV-ozone for 20 minutes without additional cleaning. A self-assembled monolayer (SAM) solution (0.3 mg/mL in absolute ethanol) was spin-coated onto the ITO substrates at 3000 rpm for 30 seconds and then annealed at 100°C for 10 minutes. The perovskite precursor solution was stirred at 60°C for 12 hours and filtered using a 0.22 μm filter. For the perovskite layer, 100 μL of the prepared perovskite solution was spin-coated onto the SAM-coated substrates at 1000 rpm for 10 seconds (ramp rate: 500 rpm/s) followed by 5000 rpm for 30 seconds (ramp rate: 2000 rpm/s). During the last 10 seconds of the spin-coating process, 200 μL of chlorobenzene was drop-cast onto the spinning film. The substrates were then annealed on a hot plate at 100°C for 40 minutes in a nitrogen atmosphere. To complete the solar cells, layers of C_{60} (40 nm), BCP (10 nm), and Ag (110 nm) were thermally evaporated.

As shown in Fig. S29, the cross-sectional images of HPDA-modified perovskite films showed vertical oriented neat grains, while the pristine film showed irregular and smaller grains with numerous grain boundaries (GBs). Due to the strong coordination effect between HPDA and Pb^{2+} , HPDA may weaken the coordination interaction between PbI_2 and DMF (Advanced Materials 2022, 34, 2200705), delaying the crystallization process, resulting in larger grain. Therefore, the optimized HPDA-modified device exhibited high PCE of 25.54%, while the champion PCE of control device was 23.69%.

Fig. S29. (a) Cross-sectional SEM images of control and HPDA-modified devices. (b) J–V curves of the champion inverted p-i-n devices with and without HPDA.

Revision: We have added the J–V curves of the champion p-i-n devices with and without HPDA as Fig. S29 in the revised supporting information file. Also, we have added the description in the revised manuscript: “We further use 1-step antisolvent method to prepare inverted PSCs and obtain a PCE of 25.54%, demonstrating that incorporating the HPDA polymer into the perovskite is a universal method (Fig. S29).”

(8) There are some typos in manuscript. For example “spieces relasing” in line 61 and “bondaling” in line 72.

Author reply: We are sorry for the typos. As suggested, we have thoroughly checked and corrected the typos we found in the revised manuscript.

Specific reply to Reviewer's comments

Title: **Boosting Mechanical Durability under High Humidity by Bioinspired Multisite Polymer for High-Efficiency Flexible Perovskite Solar Cells**

We sincerely thank the reviewers for their time, effort and valuable comments. Point-to-point reply to the comment is listed below in blue text and corresponding revision is made in the revised manuscript and provided below in green text.

Reviewer 3:

The revised manuscript by Li et al. is much improved. The addition of the GIWAXS, in situ absorption, and HRTEM analysis have clarified the role of HDPDA and its interaction with the perovskite. Generalization of their strategy to the 1-step pin devices has also elevated the impact of this manuscript. Overall, I support the publication of this revised manuscript. I have a few remaining questions:

Author reply: We sincerely thank the reviewer for recognizing the significance of our study and approving our manuscript for publication. We have carefully addressed all the provided comments, and we believe that the revisions have resulted in a substantially improved manuscript.

(1) In the fig. 2D TEM image, what is the bright contrast layer between spiro and the perovskite surface? I guess it is most likely some passivation layer. Yet, the Materials and Methods section has no mention of a passivation layer.

Author reply: We thank the reviewer for this valuable comment. The layer with bright contrast is spiro-OMeTAD layer, similar results were shown in previous literature (Nat Energy., 2019, 4, 150, J. Mater. Chem. A., 2019,7, 11867). The perovskite film is uneven and some of the spiro-OMeTAD layer is mixed with perovskite film, resulting in a blurring interface between spiro-OMeTAD and perovskite. The PSC device stacks for TEM characterization is firstly cut by focus ion beam and then further thinned by the ion beam. The spiro-OMeTAD layer is with only light element. Therefore, it presented a brighter contrast in TEM.

Fig. 2d. Cross-sectional TEM images of HPDA-modified PSCs.

(2) For the added ToF-SIMS results, how did the authors assign C₆H₄O₂ as the marking indicator for HDPDA? C₆H₄O₂ does not correspond to the chemical formula of dopamine.

Author reply: We thank the reviewer for this valuable comment. Fragment ions from organic molecules were usually used in TOF-SIMS characterizations (Fig. R1). The measurements were performed in positive ion modes. $C_6H_4O_2^+$ cluster fragments were clearly shown in the resulting spectra. From the schematics of molecular structure, $C_6H_4O_2^+$ were part of the HPDA polymer, which are sputtered after high energy ion bombardment. Moreover, we provided CHO_2^+ ions as another indicator for HPDA, and the distribution of CHO_2^+ coincided with $C_6H_4O_2^+$, also confirming that the HPDA polymer is concentrated on both the top and bottom surfaces of the perovskite layer.

Fig. R1. The breaking position of chemical bonds and the structural formula of molecular fragments during ionization process.

Fig. S21. Time of Flight Secondary Ion Mass Spectrometry (ToF-SMIS) depth profiles of the HPDA-modified PSCs.

Revision: In the revised manuscript, we have updated the time of flight secondary-ion mass spectroscopy (ToF-SIMS) of control and HPDA-modified perovskite as Fig. S21 in the supporting information.

(3) Details of the cross section TEM measurement must be clearly written in the Methods section. More generally, the Methods is missing a lot of critical information about the material characterization techniques (e.g. TEM, FIB, GIWAXS, in situ absorption).

Author reply and revision: We thank the reviewer for this valuable comment. In the revised manuscript, we have added the cross-section TEM test “The cross-sectional PSC lamella was FIB milled using an FEI Helios G4 CX Dual-beam FIB/SEM. With the electron beam, using secondary or backscattered electrons to identify the areas that need to be tested. Then with the ion beam a PSC lamella of that area is cut and transferred onto a special grid where the lamella is attached with platinum deposition. On the grid, the lamella is further thinned in the ion beam using an FEI Helios G4 CX Dual-beam FIB/SEM. until it is electron transparent. The grid is transferred from the FIB-

SEM to a TEM to analyse the lamella at high resolution.” in methods section.

We also added the other material characterization techniques (TEM, FIB, GIWAXS, in situ absorption, NMR and ToF-SIMS) “The distribution of HDPA was characterized by high-resolution transmission electron microscopy with a FEI Talos F200X operating on 200 kV electron gun. ¹H-NMR spectra were measured with a Bruker AVANCE III 600 instrument operating at 500 MHz at 298 K. Grazing-incidence wide-angle X-ray scattering (GIWAXS, BL14B1 beamline of the Shanghai Synchrotron Radiation Facility (SSRF) using X-ray with a wavelength of 0.6887 Å). In-situ UV-vis spectra of perovskite intermediate phases film are obtained using a homemade system with a fiber optic spectrometer (Ocean Optics, QE Pro). The time-of-flight secondary-ion mass spectrometry (ToF-SIMS) spectra was measured with TOF SIMS5, ION-TOF GmbH.” in methods section.